# Domain-Invariant Per-Frame Feature Extraction for Cross-Domain Imitation Learning with Visual Observations

## Abstract

Imitation learning (IL) enables agents to mimic expert behavior without reward signals but faces challenges in cross-domain scenarios with high-dimensional, noisy, and incomplete visual observations. To address this limitation, we propose Domain-Invariant Per-Frame Feature Extraction for Imitation Learning (DIFF-IL), a novel IL method that extracts domain-invariant features from individual frames and adapts them into sequences to isolate and replicate expert behaviors. We also introduce a frame-wise time labeling technique to segment expert behaviors by timesteps and assign rewards aligned with temporal contexts, enhancing task performance. Experiments across diverse visual environments demonstrate the effectiveness of DIFF-IL in addressing complex visual tasks.

## 1 Introduction

Imitation learning (IL) enables agents to learn complex behaviors by observing and replicating expert demonstrations without explicit reward signals. It is widely applied in robotics, autonomous driving, and healthcare. The simplest IL technique, behavior cloning (BC) (Bain & Sammut, 1995; Pomerleau, 1991; Ross et al., 2011; Torabi et al., 2018a), directly mimics expert datasets but struggles with generalization when agents deviate from training trajectories. Inverse reinforcement learning (IRL) addresses this by inferring reward functions from expert behavior, enabling more robust learning (Ng & Russell, 2000; Abbeel & Ng, 2004; Ziebart et al., 2008). Adversarial imitation learning (AIL) builds on IRL by aligning state–action distributions between learners and experts using adversarial frameworks (Finn et al., 2016; Fu et al., 2018; Ho & Ermon, 2016; Torabi et al., 2018b; Zhang et al., 2020), often with generative models such as GANs (Goodfellow et al., 2014). While effective in same-domain scenarios, these methods face challenges in cross-domain settings due to domain shifts complicating policy transfer (Ben-David et al., 2006).

In cross-domain scenarios, mismatches arise from differences in viewpoints, dynamics, embodiments, and state spaces, creating hurdles for IL applications. For instance, autonomous driving may require learning from simulations while operating in real-world environments, or robots may rely on visual data to control joints. These shifts exacerbate learning difficulties, particularly with high-dimensional and noisy visual data, where even minor variations can disrupt alignment and stability. To address these issues, cross-domain IL techniques extract domain-invariant features from visual datasets to align source and target domains while retaining task information (Li et al., 2018; Liu et al., 2018; Cetin & Celiktutan, 2021; Shang & Ryoo, 2021). By focusing on features independent of domain-specific factors, these methods enable learners to mimic expert behavior using visual demonstrations, improving IL's effectiveness across diverse real-world scenarios (Sermanet et al., 2018).

Existing IL methods often rely on image sequences spanning multiple timesteps to identify domain-invariant features for IRL and reward design, as single images cannot fully capture an agent's evolving behavior. However, these approaches frequently struggle with the complexity of sequence spaces, leading to misaligned features, poorly designed rewards, and suboptimal imitation of expert policies. To address these challenges, we propose Domain-Invariant Per-Frame Feature Extraction for Imitation Learning (DIFF-IL). DIFF-IL introduces two key contributions: (1) per-frame domain-invariant feature extraction to robustly isolate domain-independent task-relevant behaviors, and (2) frame-wise time labeling to segment expert behaviors by timesteps and assign rewards based on temporal alignment. Together, these innovations enable precise domain alignment and effective imitation, even in scenarios with limited overlap between source domain data and expert actions.

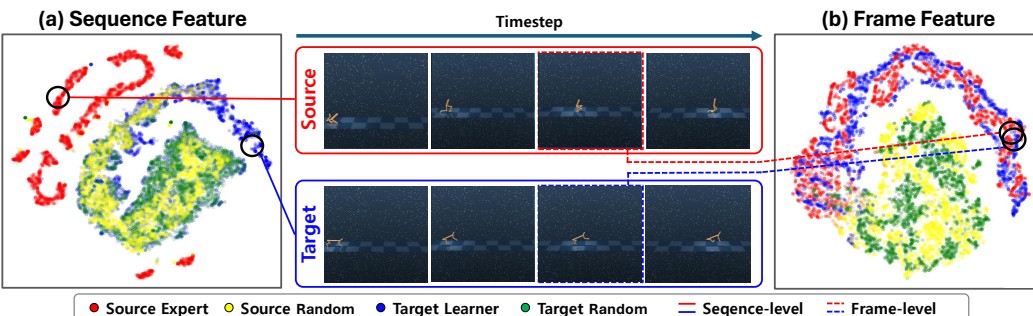

Figure 1: t-SNE visualization of features: (a) sequence-based IL methods, (b) DIFF-IL (ours)

Fig. 1 highlights the strengths of the proposed DIFF-IL method in the Walker (source)-to-Cheetah (target) environment. In this scenario, the Cheetah agent (target learner) aims to move forward quickly by mimicking expert demonstrations from Walker agents (source expert), despite differing dynamics. Fig. 1(a) presents a t-SNE visualization of latent features from sequences of four-frame sequences extracted using sequence-based IL methods. While source expert and target learner agents share similar positions and speeds, their features fail to align due to residual domain-specific information, leading to inaccurate rewards and suboptimal learning. In contrast, Fig. 1(b) shows that DIFF-IL seamlessly aligns latent features of individual image frames across domains, effectively removing domain-specific artifacts while preserving expertise-critical details. This enables DIFF-IL to extract truly domain-invariant features, allowing the learner to accurately mimic expert behaviors, unlike sequence-based methods that fail to achieve robust domain adaptation. Moreover, when source expert and random behaviors overlap minimally, traditional methods often misclassify slight deviations as expert behavior, hindering learning. DIFF-IL addresses this by incorporating frame-wise time labeling, which segments expert behaviors into finer temporal contexts. By assigning higher rewards to frames closer to the goal, DIFF-IL guides the agent to progressively replicate expert trajectories, ensuring robust alignment and successful task completion under challenging conditions.

## 2 RELATED WORKS

**Imitation Learning:** IL trains agents to mimic expert behaviors. Behavior cloning uses supervised learning for replication (Kelly et al., 2019; Sasaki & Yamashina, 2020; Reddy et al.; Florence et al., 2022; Shafiullah et al., 2022; Hoque et al., 2023; Li et al., 2024; Mehta et al., 2025), while Inverse RL derives reward functions from expert demonstrations (Abbeel & Ng, 2004; Ziebart et al., 2008; Dadashi et al., 2020; Wang et al., 2022). Building on IRL, adversarial methods distinguish between expert and learner behaviors to provide reward signals (Ho & Ermon, 2016; Fu et al., 2017; Li et al., 2017; Peng et al., 2018; Lee et al., 2019; Ghasemipour et al., 2020). There are also approaches that aim to integrate the strengths of BC and IRL (Watson et al., 2024). Offline IL methods enable robust training without interaction (Kim et al., 2022; Xu et al., 2022; Ma et al., 2022; Hong et al., 2023; Yan et al., 2023; Li et al., 2023; Zhang et al., 2023; Sun et al., 2023), and strategies addressing dynamic shifts through diverse tasks have also been proposed (Chae et al., 2022).

**Cross-Domain Imitation Learning (CDIL):** CDIL transfers expert behaviors across domains with differences in perspectives, dynamics, or morphologies. Approaches include using the Gromov-Wasserstein metric for cross-domain similarity rewards (Fickinger et al., 2022), timestep alignment (Sermanet et al., 2018; Liu et al., 2018; Kim et al., 2020; Raychaudhuri et al., 2021), and temporal cycle consistency to address alignment issues (Zakka et al., 2022). Techniques also involve removing domain-specific information via mutual information (Cetin & Celiktutan, 2021), maximizing transition similarity (Franzmeyer et al., 2022), or combining cycle consistency with mutual information (Yin et al., 2022). Adversarial networks and disentanglement strategies further enhance domain invariance (Stadie et al., 2017; Sharma et al., 2019; Shang & Ryoo, 2021; Choi et al., 2024).

**Imitation from Observation (IfO):** IfO focuses on learning behaviors without action information. Approaches can be divided into those leveraging vectorized observations provided by the environment (Torabi et al., 2018b; Zhu et al., 2020; Desai et al., 2020; Gangwani et al., 2022; Chang et al., 2022; Liu et al., 2023; Freund et al., 2023) and those utilizing images to model behaviors (Li et al., 2018; Liang et al., 2018; Das et al., 2021; Karnan et al., 2022b;a; Belkhale et al., 2023; Zhang et al., 2024; Xie et al., 2024; Ishida et al., 2024; Aoki et al., 2024). Image-based methods have gained attention for enabling robots to learn from human behavior captured in images, facilitating tasks like mimicking human actions(Sheng et al., 2014; Yu et al., 2018; Zhang et al., 2022; Mandlekar et al., 2023).

## 3 BACKGROUND

### 3.1 MARKOV DECISION PROCESS AND RL SETUP

In this paper, all environments are modeled as a Markov Decision Process (MDP) defined by the tuple $\mathcal{M} = (\mathcal{S}, \mathcal{A}, P, R, \gamma, \rho_0)$, where $\mathcal{S}$ is the state space, $\mathcal{A}$ the action space, $P : \mathcal{S} \times \mathcal{A} \times \mathcal{S} \to \mathbb{R}^+$ the state transition probability, $R : \mathcal{S} \times \mathcal{A} \to \mathbb{R}$ the reward function, $\gamma \in (0, 1)$ the discount factor, and $\rho_0$ the initial state distribution. At each timestep $t$, the agent selects an action $a_t \sim \pi$ from a stochastic policy $\pi : \mathcal{S} \times \mathcal{A} \to \mathbb{R}^+$. The environment provides a reward $r_t = R(s_t, a_t)$ and the next state $s_{t+1} \sim P(\cdot|s_t, a_t)$. The goal in reinforcement learning (RL) is to optimize the policy $\pi$ to maximize the discounted cumulative reward $\sum_t \gamma^t r_t$.

### 3.2 ADVERSARIAL IMITATION LEARNING

IL trains a learner policy $\pi^L$ to mimic an expert policy $\pi^E$ using an offline dataset $\mathcal{B}^E$ of expert trajectories $\tau^{\pi^E}$, where each trajectory $\tau^\pi := (s_0, a_0, s_1, a_1, \cdots, s_H)$ consists of state-action pairs, with $a_t \sim \pi(\cdot|s_t)$ for $t = 0, \cdots, H - 1$, and $H$ is the episode length. To improve IL performance, Generative Adversarial IL (GAIL) (Ho & Ermon, 2016) applies GAN (Goodfellow et al., 2014) principles to IL by using a label discriminator $F$ to distinguish between learner trajectories $\tau^{\pi^L}$ (label 0) and expert trajectories $\tau^{\pi^E}$ (label 1). Rewards are defined in an IRL framework where $F$ assigns higher values to actions harder to classify. The learner $\pi^L$ acts as a generator, confusing $F$ by maximizing these rewards through online RL, thereby aligning its trajectory distribution with $\pi^E$ via adversarial training. Building on this, Adversarial IRL (AIRL) (Fu et al., 2017) introduces a reward structure that further enhances the learner's ability to perform IL, as follows:

$$R_F(s_t, a_t, s_{t+1}) = \log F(s_t, a_t, s_{t+1}) - \log(1 - F(s_t, a_t, s_{t+1})) \tag{1}$$

### 3.3 CROSS-DOMAIN IL WITH VISUAL OBSERVATIONS

To enable practical IL in cross-domain settings, the expert's environment is modeled as an MDP $\mathcal{M}^S$ and the learner's as $\mathcal{M}^T$. The goal is to train the learner by minimizing domain gaps and mimicking expert behavior via distribution matching (Torabi et al., 2018b; Gangwani et al., 2022; Liu et al., 2023). In practice, image observations from offline datasets are often used (Stadie et al., 2017; Kim et al., 2020; Zakka et al., 2022). The observation space $\mathcal{O}^d$ belongs to $\mathcal{M}^d$ for $d \in \{S, T\}$, with each image frame $o_t^d$ capturing time $t$. As single frames cannot capture dynamics, IL instead uses sequences $o_{\text{seq},t}^d = (o_{t-L+1}^d, \cdots, o_t^d)$, with $L = 4$ in this work.

Recent cross-domain IL methods leverage random policies to capture domain characteristics (Cetin & Celiktutan, 2021; Choi et al., 2024). Here, we define $\pi^{SE}$ as the source expert (SE) policy, $\pi^{SR}$ as the source random (SR) policy, $\pi^{TL}$ as the target learner (TL) policy, and $\pi^{TR}$ as the target random (TR) policy. Offline datasets $\mathcal{B}^\pi$, consisting of visual demonstration trajectories $\tilde{\tau}^\pi = (o_0^d, a_0^d, \cdots, o_H^d)$ generated by $\pi \in \{\pi^{SE}, \pi^{SR}, \pi^{TR}\}$, are provided. For simplicity, we denote $\mathcal{B}^{SE}$, $\mathcal{B}^{SR}$, and $\mathcal{B}^{TR}$ as the datasets corresponding to their respective policies. Using these datasets, which lack access to true states, $\pi^{TL}$ is trained to mimic expert behavior in the target domain by reducing the domain gap.

## 4 METHODOLOGY

### 4.1 DOMAIN-INVARIANT PER-FRAME FEATURE EXTRACTION

In this section, we propose a domain-invariant per-frame feature extraction (DIFF) method to eliminate domain-specific information while preserving expertise-related details, enabling effective domain adaptation prior to utilizing image sequences for expertise assessment. Specifically, we define a shared encoder $p$ and domain-specific decoders $q^d$, where $d \in \{S, T\}$. The encoder $p$ encodes image data into latent features $z_t^d \sim p(\cdot|o_t^d)$, while each decoder $q^d$ reconstructs the original image as $\hat{o}_t^d = q^d(z_t^d)$, ensuring $z_t^d$ captures essential image characteristics. However, $z_t^d$ may still contain irrelevant domain-specific details (e.g., background, camera angles) in addition to expertise-related information (e.g., agent position, joint angles), hindering the learner's ability to interpret expertise.

To mitigate residual domain-specific details in latent features $z_t^d$, we employ a Wasserstein GAN (WGAN) (Gulrajani et al., 2017b), with the encoder $p$ as the generator and a frame discriminator $D_f$ to classify whether $z_t^d$ originates from the source or target domain. The discriminator $D_f$ is trained to distinguish domains, while the encoder $p$ attempts to confuse $D_f$, aligning $z_t^d$ distributions across domains to remove domain-specific information. Meanwhile, the encoder-decoder structure preserves

task-relevant information by minimizing reconstruction errors. A consistency loss inspired by Zhu et al. (2017) further ensures that $z_t^d$ retains expertise-related information even after cross-domain transfer (Choi et al., 2024). Specifically, when $z_t^d$ is processed through the opposite domain's decoder $q^{d'}$ and re-encoded by $p$, the resulting latent $\hat{z}_t^d \sim p(\cdot|q^{d'}(z_t^d))$ remains consistent, maintaining task relevance while removing artifacts. Training involves three components: frame discriminator loss $\mathcal{L}_{\text{disc},f}(D_f)$, frame generator loss $\mathcal{L}_{\text{gen},f}(p)$, and encoder-decoder loss $\mathcal{L}_{\text{enc-dec}}(p,q)$, defined as:

$$\mathcal{L}_{\text{disc},f} := \mathbb{E}_{z_t^S \sim p(\cdot|o_t^S),\, z_t^T \sim p(\cdot|o_t^T)} \left[ -D_f(z_t^S) + D_f(z_t^T) \right] + \lambda_{\text{gp},f} \cdot GP,$$

$$\mathcal{L}_{\text{gen},f} := \mathbb{E}_{z_t^S \sim p(\cdot|o_t^S),\, z_t^T \sim p(\cdot|o_t^T)} \left[ D_f(z_t^S) - D_f(z_t^T) \right], \qquad (2)$$

$$\mathcal{L}_{\text{enc-dec}} := \sum_{d=S,T} \mathbb{E}_{z_t^d \sim p(\cdot|o_t^d)} \Big[ \underbrace{\|o_t^d - \hat{o}_t^d\|_2}_{\text{Reconstruction Loss}} + \underbrace{\|\bar{z}_t - \hat{z}_t^d\|_2}_{\text{Feature Consistency Loss}} \Big]$$

where $\hat{o}_t^d = q^d(z_t^d)$, $\hat{z}_t^d \sim p(\cdot|q^{d'}(z_t^d))$, and $\bar{x}$ represents stopping gradient flow for $x$ and $GP$ represents the Gradient Penalty term to guarantee stable learning. Samples are drawn from domain-specific buffers $\mathcal{B}^S := \mathcal{B}^{SR} \cup \mathcal{B}^{SE}$ and $\mathcal{B}^T := \mathcal{B}^{TR} \cup \mathcal{B}^{TL}$, where $\mathcal{B}^{TL}$ stores trajectories from $\pi^{TL}$ during training. This approach ensures domain-invariant features while preserving task-relevant information, enabling robust cross-domain expertise alignment.

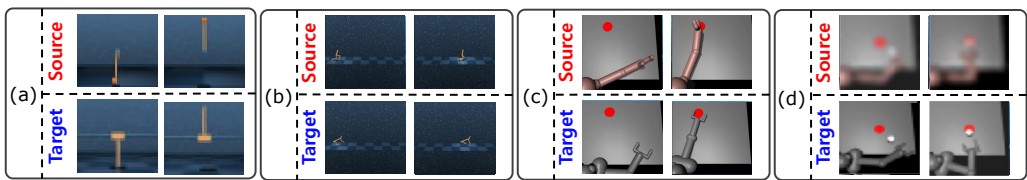

Figure 2: Image mappings of DIFF-IL based on aligned latent features in (a) Pendulum, and (b) MuJoCo tasks, (c) Robot Manipulation tasks (d) Robot Manipulation tasks with resolution shifts

Fig. 2 showcases image mappings with the proposed DIFF, aligning source and target images based on their closest latent features $z_t^d$, $d \in \{S,T\}$ across (a) Pendulum, swinging up and balancing upright; (b) MuJoCo, moving forward rapidly despite differing dynamics; (c) Robot Manipulation tasks using high-DoF robotic agents performing pushing and reaching behaviors, designed to better reflect real-world manipulation scenarios, and (d) Robot Manipulation tasks with resolution shifts, where source-domain images are intentionally downsampled to simulate deployment conditions with lower visual fidelity before being matched to higher-resolution target-domain images. Each environment uses distinct agents in the source and target domains but shares the same underlying task objective. Across all task categories, the resulting feature representations exhibit near one-to-one alignment, capturing task-relevant details such as pole angles, agent positions, and object locations while minimizing domain-specific differences such as appearance, physics, and image resolution, demonstrating the method's ability to preserve expertise-related information for accurate imitation.

## 4.2 Sequential Matching with Expertise Labeling

The proposed per-frame feature extraction removes domain-specific information from individual frames. Building on this, we utilize sequences of these features, as in existing AIL methods, for expertise assessment. At each time step $t$, the feature sequence is defined as $z_{\text{seq},t}^d := z_{t-L+1}^d, \cdots, z_t^d$, with $L$ as the fixed sequence length. To classify expertise, we introduce a sequence label discriminator $F_{\text{label},s}(z_{\text{seq},t}^d) \in [0,1]$, trained to label feature sequences $z_{\text{seq},t}^d$ from $\mathcal{B}^{SE}$ as expert (label 1) and others as non-expert (label 0). Although per-frame domain-specific information is removed, domain-specific sequence differences (e.g., speeds, step sizes) may persist. To address this, we extend WGAN to feature sequences using a sequence discriminator $D_s$, ensuring residual sequence-related domain-specific information is further eliminated. In summary, the training for feature sequences also includes three components: sequence discriminator loss $\mathcal{L}_{\text{disc},s}(D_s)$, sequence generator loss $\mathcal{L}_{\text{gen},s}(p)$, and sequence label loss $\mathcal{L}_{\text{label},s}(F_{\text{label},s}, p)$, are defined as:

$$\mathcal{L}_{\text{disc},s} := \mathbb{E}_{z_{\text{seq},t}^S \sim p(\cdot|o_{\text{seq},t}^S),\, z_{\text{seq},t}^T \sim p(\cdot|o_{\text{seq},t}^T)} \left[ -D_s(z_{\text{seq},t}^S) + D_s(z_{\text{seq},t}^T) \right] + \lambda_{\text{gp},s} \cdot GP,$$

$$\mathcal{L}_{\text{gen},s} := \mathbb{E}_{z_{\text{seq},t}^S \sim p(\cdot|o_{\text{seq},t}^S),\, z_{\text{seq},t}^T \sim p(\cdot|o_{\text{seq},t}^T)} \left[ D_s(z_{\text{seq},t}^S) - D_s(z_{\text{seq},t}^T) \right], \qquad (3)$$

$$\mathcal{L}_{\text{label},s} := \sum_{d=S,T} \mathbb{E}_{z_{\text{seq},t}^d \sim p} \Big[ \underbrace{\text{BCE}(F_{\text{label},s}(z_{\text{seq},t}^d),\ \mathbb{1}_{o_{\text{seq},t}^d \sim \mathcal{B}^{SE}})}_{\text{Label Loss}} \Big],$$

where BCE is Binary Cross Entropy and $\mathbb{1}_x$ the indicator function, 1 if condition $x$ is true and 0 otherwise. The loss ensures that the feature sequence $z^d_{\text{seq},t}$ is free from domain-specific information, enabling pure expertise assessment. For WGAN, to balance per-frame and sequence mappings, the unified WGAN loss is redefined as: $\mathcal{L}_{\text{WGAN}} = \lambda_{\text{disc}}\mathcal{L}_{\text{disc}} + \lambda_{\text{gen}}\mathcal{L}_{\text{disc}}$, where $\lambda_{\text{disc}}$ and $\lambda_{\text{gen}}$ are scaling coefficients for discriminator and generator losses. The losses are defined as:

$$\mathcal{L}_{\text{disc}} := \alpha\mathcal{L}_{\text{disc},f} + (1-\alpha)\mathcal{L}_{\text{disc},s}$$
$$\mathcal{L}_{\text{gen}} := \alpha\mathcal{L}_{\text{gen},f} + (1-\alpha)\mathcal{L}_{\text{gen},s}, \tag{4}$$

where $\alpha \in (0,1)$ is the WGAN control parameter, adjusting the balance between per-frame WGAN ($\mathcal{L}_{\text{disc},f}$, $\mathcal{L}_{\text{gen},f}$) and sequence WGAN ($\mathcal{L}_{\text{disc},s}$, $\mathcal{L}_{\text{gen},s}$). Due to the large number of losses, most scales are fixed, while parameter search focuses on key WGAN hyperparameters $\alpha$, $\lambda_{\text{disc}}$, and $\lambda_{\text{gen}}$, most relevant to the proposed DIFF method. Details on other loss scales are provided in Appendix B.

### 4.3 FRAME-WISE TIME LABELING AND REWARD DESIGN

The trained $F_{\text{label},s}$ evaluates the expertise of feature sequences, with labels influenced by overlap between the source domain's expert data $\mathcal{B}^{SE}$ and random data $\mathcal{B}^{SR}$ during sequence label loss training. When expert and random sequences overlap significantly, expert labels are distributed between 0 and 1, helping the target learner mimic critical behaviors effectively. However, as shown in Fig. 1(b), minimal overlap results in most expert data being labeled as 1 when slightly deviating from random data, making important behaviors harder to identify. To address this, we propose a frame-wise time labeling method, which segments expert behavior by timesteps and guides the learner to prioritize later frames to achieve task objectives. To implement the frame label, we define a frame label discriminator $F_{\text{label},f}(z^d_t) \in [0,1]$, trained with the frame label loss $\mathcal{L}_{\text{label},f}(F_{\text{label},f})$:

$$\mathcal{L}_{\text{label},f} := \mathbb{E}_{z^S_t \sim p(\cdot|o^S_t)}\left[\text{BCE}\left(F_{\text{label},f}(z^S_t), y_t\right)\right], \tag{5}$$

where $y_t$, the time label for $o^S_t$ at time $t$, is defined as:

$$y_t = \begin{cases} \left(\frac{t}{H_{\tilde{\tau}}} + 1\right)/2 & \text{if } o^S_t \sim \mathcal{B}^{SE}, \\ 0 & \text{otherwise.} \end{cases}$$

Here, $H_{\tilde{\tau}}$ denotes the episode length of $\tilde{\tau} \in \mathcal{B}^{SE}$. Time labeling is trained solely on the source domain, since expert data is unavailable in the target. Unlike prior methods aligning features by timesteps (Sermanet et al., 2018), our approach segments expert behavior with labels for finer expertise granularity. The frame label discriminator assigns higher labels to later timesteps, guiding the learner to replicate actions aligned with task objectives. Fig. 3 illustrates how time labeling $y_t$ prioritizes later-stage frames, emphasizing temporal progression of expertise and ensuring precise replication of expert behaviors for accurate task completion.

In summary, we propose DIFF for IL (DIFF-IL), which integrates the domain-invariant per-frame feature extraction with AIL principles from Section 3. DIFF-IL leverages sequence labels via $F_{\text{label},s}$ to guide the learner in mimicking expert behavior, while frame-wise time labeling through $F_{\text{label},f}$ emphasizes later-stage frames, prioritizing the temporal progression of expertise. DIFF-IL integrates sequence and frame labels into a reward function to maximize alignment for accurate replication:

$$\hat{R}_t = -\log(1 - F_{\text{label},s}(z^T_{\text{seq},t+1}) \cdot F_{\text{label},f}(z^T_{t+1})), \tag{6}$$

where $z^T_{\text{seq},t+1} \sim p(\cdot|o^T_{\text{seq},t+1})$, $z^T_{t+1} \sim p(\cdot|o^T_{t+1})$ for $o^T_{\text{seq},t+1}, o^T_{t+1} \sim \mathcal{B}^{TL}$, and observations at time $t+1$ are used in $\hat{R}_t$ to capture the effect of action $a_t$. This reward adopts only the positive part of AIRL's design in Section 3 since it remains effective in maximizing labels. For implementation, the target learner $\pi^{TL}$ aims to maximize the reward sum $\sum_t \gamma^t \hat{R}_t$ using the Soft Actor-Critic (SAC) Haarnoja et al. (2018), a widely used RL method that exploits entropy for exploration. Each iteration includes $N_{\text{model,train}}$ model training steps and $N_{\text{RL,train}}$ RL training steps, with updates to $p$, $q^S$, $q^T$, $F_{\text{label},f}$, and $F_{\text{label},s}$ every $n$ periods. The overall structure of DIFF-IL is illustrated in Fig. 4, with additional implementation details, losses and the complete algorithm, provided in Appendix B.

### 5 EXPERIMENTS

We evaluate DIFF-IL against various cross-domain IL methods on DeepMind Control Suite (DMC) (Tassa et al., 2018) and MuJoCo (Todorov et al., 2012a), pairing similar tasks as source and target

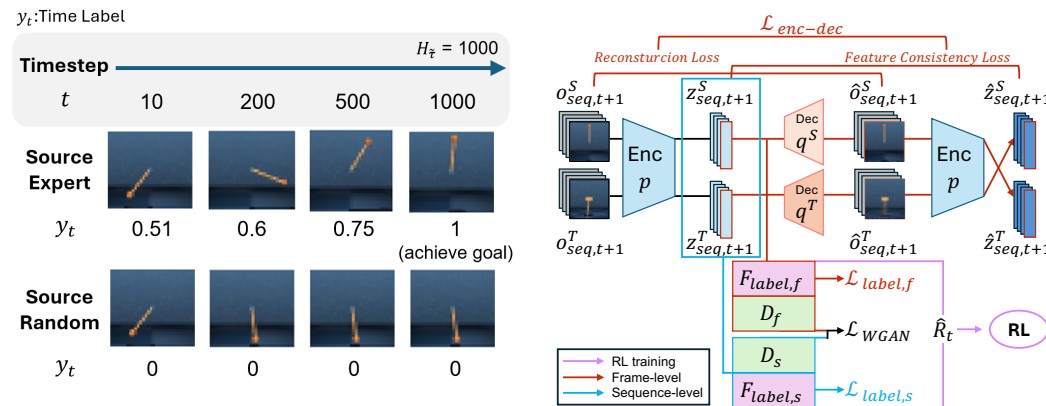

Figure 3: Illustration of frame-wise time labeling

Figure 4: Structure of the proposed DIFF-IL

domains. The evaluation shows how well the target learner mimics the source expert and analyze the effectiveness of key components.

## 5.1 EXPERIMENTAL SETUP

For comparison, we evaluate cross-domain IL methods using images: **TPIL** (Stadie et al., 2017), which extracts domain-invariant features from image sequences; **DeGAIL** (Cetin & Celiktutan, 2021), which enhances domain information removal with mutual information; **D3IL** (Choi et al., 2024), which isolates expertise-related behavior using dual consistency loss; and **DIFF-IL** (Ours). Additionally, **GWIL** (Fickinger et al., 2022), a state-based approach leveraging Gromov-Wasserstein distance, serves as a baseline. For DIFF-IL, we primarily tuned WGAN hyperparameters ($\alpha$, $\lambda_{\text{disc}}$, $\lambda_{\text{gen}}$), fixing other loss scales. $\alpha = 0.5$ delivered consistently strong performance across environments, while $\lambda_{\text{disc}}$ and $\lambda_{\text{gen}}$ were optimized per environment. Detailed descriptions of each algorithm, along with our hyperparameter setup, are provided in Appendix C.

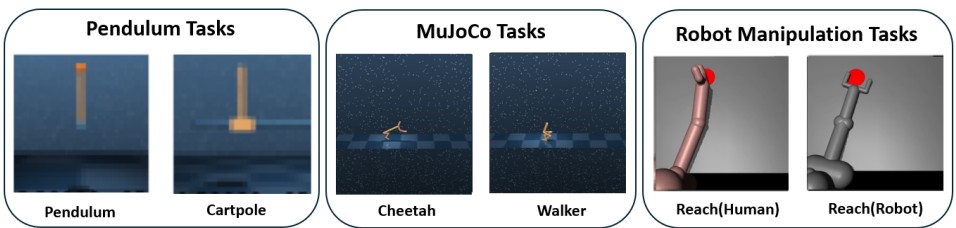

Figure 5: Examples of environments used in our experiments.

## 5.2 ENVIRONMENTAL SETUP

We evaluate baselines under substantial cross-domain shifts, focusing on adaptation across tasks that differ in agent morphology and control complexity (e.g., number of joints, action dimensionality, dynamics) rather than superficial factors such as viewpoint or color. Cross-domain scenarios are denoted as *A-to-B*, where *A* and *B* indicate the source and target domains. All environments are implemented in MuJoCo (Todorov et al., 2012b) and grouped into three categories: **Pendulum Tasks**, **MuJoCo Tasks**, and **Robot Manipulation Tasks**. Pendulum and MuJoCo locomotion suites are challenging benchmarks that stress morphology and dynamics shifts, while the Robot Manipulation suite is specifically designed to probe scalability toward real-world deployment by introducing more challenging settings with high-DoF robotic agents and additional visual fidelity variations.

**Pendulum Tasks** Pendulum tasks involve controlling pole agents to maintain balance or reach targets, as shown in Fig. 5, and are grouped into three categories: Inverted Pendulum Tasks, including Inverted Pendulum (IP) with a single pole and Inverted Double Pendulum (IDP) with two interconnected poles, focusing on vertical balance with rewards increasing as poles approach an upright position; Reacher Tasks, where Reacher2 (RE2) and Reacher3 (RE3) involve two- and three-joint robotic arms reaching one of 16 targets, with rewards reflecting the negative distance to the target; and DMC Pendulum Tasks, including Cartpole Swingup (CS), Pendulum (Pend), and Acrobot, emphasizing pole balance with rewards increasing for upright positions.

**MuJoCo Tasks** These tasks use Walker, Cheetah, and Hopper locomotion agents aiming to move as quickly as possible. The camera is adjusted to capture movements, with rewards solely on forward

speed, increasing as the agents move faster. Each agent presents distinct locomotion challenges: Walker is a bipedal agent requiring stable coordination, Cheetah is a quadrupedal agent optimized for high-speed running, and Hopper is a single-legged agent that must balance and hop forward.

**Robot Manipulation Tasks**    To evaluate scalability toward real-world scenarios, we introduce Robot Manipulation tasks involving high-DoF robotic agents performing pushing and reaching behaviors under completely distinct embodiments. This setup considers cross-domain transfer between a simple two-finger gripper (Robot) and a dexterous multi-fingered hand (Humanoid), reflecting the challenging embodiment gaps often encountered in practice. To further approximate sim-to-real conditions, we also include resolution-shifted variants where source images are downsampled and then upsampled to simulate low-fidelity visual sensors, while the target domain retains high-resolution images. These settings jointly evaluate the robustness of distribution matching under both embodiment and visual fidelity gaps. Concretely, we study two task families: Pusher, where the agent must move an object to a target location, and Reach, where the end-effector must reach a designated goal position.

To evaluate domain adaptation, we define 8 pendulum task scenarios (Pend-to-CS, Pend-to-Acrobot, CS-to-Pend, CS-to-Acrobot, RE3-to-RE2, RE2-to-RE3, IP-to-IDP, and IDP-to-IP), 6 MuJoCo task scenarios (Walker-to-Cheetah, Walker-to-Hopper, Cheetah-to-Walker, Cheetah-to-Hopper, Hopper-to-Cheetah, and Hopper-to-Walker), and 8 robot manipulation scenarios. The manipulation set consists of four base transfers (Pusher:R-to-H, Pusher:H-to-Robot, Reach:R-to-H, Reach:H-to-R) and their resolution shifted counterparts denoted by '-Res'. For Robot Manipulation tasks, the suffix 'R' indicates the Robot agent, while the suffix 'H' represents the Humanoid agent. The Acrobot task, due to its complexity, is used only as a target environment. Resolution shifted variants change only the observation resolution and are used to isolate the effect of visual fidelity on cross domain alignment. In each scenario, an expert policy is trained using SAC in the source environment to construct the source expert dataset. Performance is measured as the return achieved by the target learner in the target environment, averaged over 5 random seeds, with results reported as means and standard deviations (shaded areas in graphs and $\pm$ values in tables). Additional details on the environments and offline data construction are provided in the Appendix C.3.

| Tasks | | Average Return | | | | |
|---|---|---|---|---|---|---|
| | | **DIFF-IL (Ours)** | **D3IL** | **DeGAIL** | **GWIL** | **TPIL** |
| **Pendulum** | IP-to-IDP | **9358.51 ± 0.86** | 9300.2 ± 271.4 | 479.4 ± 173.1 | 461.79 ± 64.80 | 174.52 ± 52.30 |
| | IDP-to-IP | **1000.00 ± 0.00** | **1000.00 ± 0.0** | 27.00 ± 192.64 | 417.03 ± 60.00 | 11.07 ± 5.84 |
| | RE3-to-RE2 | -3.33 ± 0.80 | **-3.16 ± 0.73** | -6.43 ± 0.70 | -11.57 ± 0.30 | -9.71 ± 1.93 |
| | RE2-to-RE3 | **-2.27 ± 0.56** | -3.99 ± 1.35 | -10.05 ± 1.24 | -9.84 ± 0.12 | -10.09 ± 1.91 |
| | Pend-to-CS | **739.51 ± 48.54** | 528.54 ± 106.5 | 1.65 ± 1.02 | 0.00 ± 0.21 | 4.70 ± 14.56 |
| | Pend-to-Acrobot | **128.24 ± 40.58** | 62.51 ± 28.67 | 3.96 ± 3.16 | 6.52 ± 6.46 | 3.77 ± 4.52 |
| | CS-to-Pend | **803.50 ± 46.00** | 646.81 ±127.4 | 5.99 ± 17.48 | 0.21± 0.74 | 85.82 ± 165.01 |
| | CS-to-Acrobot | **64.86 ± 25.79** | 53.97 ± 30.36 | 2.66 ± 2.42 | 6.83 ± 7.81 | 0.54 ± 1.21 |
| **MuJoCo** | Cheetah-to-Walker | **2.84 ± 0.38** | 0.12 ± 0.07 | 0.00 ± 0.00 | -0.06 ± 0.06 | 0.00 ± 0.00 |
| | Cheetah-to-Hopper | **1.14 ± 0.19** | 0.16 ± 0.04 | -0.07 ± 0.07 | -0.02 ± 0.04 | 0.02 ±0.03 |
| | Walker-to-Cheetah | **4.54 ± 0.86** | 2.78 ± 0.61 | 0.06 ± 0.13 | -0.51 ± 0.13 | 1.38 ±0.24 |
| | Walker-to-Hopper | **1.04 ± 0.28** | 0.68 ± 0.11 | 0.00 ± 0.01 | 0.00 ± 0.03 | 0.00 ±0.03 |
| | Hopper-to-Walker | **2.01 ± 0.19** | -0.06 ± 0.02 | 0.00 ± 0.01 | -0.03 ± 0.05 | 0.00 ±0.01 |
| | Hopper-to-Cheetah | **2.32 ± 0.58** | 0.22 ± 0.22 | 0.50 ± 0.37 | 0.33 ± 0.35 | 0.00 ±0.00 |
| **Robot Manipulation** | Pusher:R-to-H | **-59.54 ± 17.36** | -70.92 ± 9.40 | -83.48 ± 22.78 | -101.2 ± 19.10 | -105.75 ±1.34 |
| | Pusher:H-to-R | **-52.79 ± 17.78** | -54.04 ± 9.75 | -105.75 ± 0 | -103.2 ± 14.16 | -77.00 ±17.98 |
| | Reach:R-to-H | **-139.18 ± 2.36** | -221.27 ± 5.61 | -323.92 ± 19.2 | -336.27 ± 54.7 | -349.92 ±58.6 |
| | Reach:H-to-R | **-137.10 ± 9.20** | -213.29 ± 16.3 | -248.87 ± 62.2 | -314.92 ± 59.6 | -324.86 ±63.7 |
| | Pusher-Res:R-to-H | **-60.43 ± 14.52** | -76.92 ± 12.11 | -95.24 ± 20.71 | -105.61 ± 1.45 | -105.55 ± 1.72 |
| | Pusher-Res:H-to-R | **-54.31 ± 16.02** | -62.04 ± 11.40 | -105.75 ± 0 | -103.8 ± 12.60 | -85.00 ± 18.10 |
| | Reach-Res:R-to-H | **-144.8 ± 10.15** | -233.60 ± 9.10 | -320.07 ± 20.5 | -353.6 ± 46.65 | -351.0 ± 52.46 |
| | Reach-Res:H-to-R | **-146.2 ± 12.70** | -231.1 ± 21.61 | -245.2 ± 70.14 | -318.9 ± 41.50 | -330.8 ± 57.71 |

Table 1: Performance comparison on Pendulum, MuJoCo, and Robot Manipulation tasks

### 5.3 PERFORMANCE COMPARISON

From the performance comparison, Table 1 reports the mean final returns, averaged over the last 10 episodes and categorized by method across all environments. The results demonstrate that the proposed algorithm consistently outperforms existing cross domain IL methods across diverse tasks. In particular, DIFF-IL achieves significantly higher returns in challenging settings like Walker-to-Cheetah and Robot-to-Humanoid, where domain shifts involve substantial changes in both dynamics

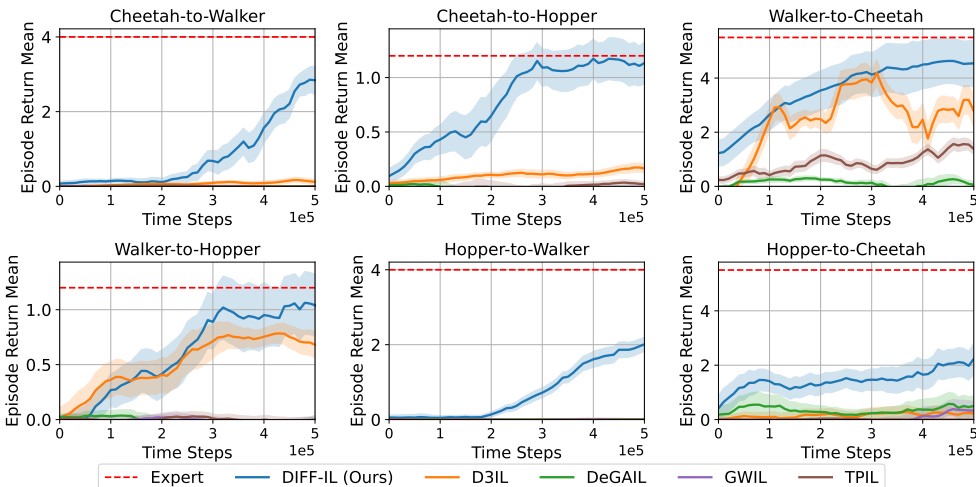

Figure 6: Performance comparison: Learning curves on MuJoCo tasks

and morphology, effectively adapting expert behaviors. Fig. 6 presents learning curves in MuJoCo environments, with additional curves for Pendulum and Robot Manipulation in Appendix D. The visualizations reveal that DIFF-IL closely mimics expert trajectories with smoother transitions and higher scores, whereas competing methods often fail to replicate expert performance, and in Robot Manipulation it maintains clear margins across both embodiments and sensing conditions, indicating stable visuomotor alignment that translates beyond simulation locomotion. However, when Hopper is the source, the target performance plateaus below expert levels due to physical speed limitations inherited during adaptation, reflecting intrinsic agent constraints. Overall, the results show superior performance across most tasks and faster convergence than other methods, attributable to a well structured reward design enabling effective trajectory alignment and stable policy learning.

We also compared DIFF-IL with Time-Contrastive Networks (TCN), another IL method that, although not image-based, uses time-labeling to align temporally similar frames. Across representative tasks, DIFF-IL consistently achieved substantially higher performance, highlighting the advantage of combining distribution matching with temporal guidance (see Appendix G). To further assess practicality, we conducted two additional studies reported in Appendix D. In the limited data evaluation, DIFF-IL trained with only 10% of the expert data still delivered strong performance, demonstrating robustness under data scarcity. In the complexity comparison, DIFF-IL required only about 10% more memory than D3IL but trained nearly three times faster per epoch, underscoring both its efficiency and scalability. Together, these additional experiments confirm that DIFF-IL is both robust to data limitations and computationally efficient, making it a practical solution for cross-domain IL.

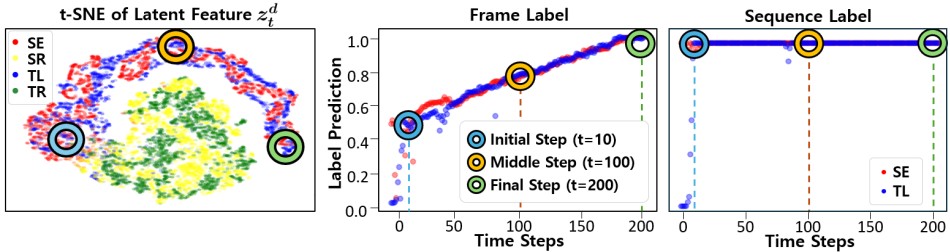

Figure 7: Latent Feature t-SNE and timestep-align label estimations on Walker-to-Cheetah tasks

## 5.4 TRAJECTORY ANALYSIS OF DOMAIN TRANSFER IN DIFF-IL

In IL, understanding how expert behavior is mimicked is as crucial as performance. To analyze how the proposed DIFF-IL effectively transfers a source expert's behavior across domains, Fig. 7 focuses on the Walker-to-Cheetah environment, where DIFF-IL significantly outperforms other methods. The figure presents a t-SNE visualization of latent features from the source expert (SE), source random (SR), target learner (TL), and target random (TR), identical in format to Fig. 1. This visualization confirms that the learned TL features align closely with those of SE while excluding domain-specific artifacts, effectively capturing task-relevant states.

The t-SNE plot also reveals minimal overlap between SE and SR data distributions. Consequently, methods relying solely on sequence labels often misclassify behaviors slightly deviating from random policies as expert, leading to suboptimal mimicry. In contrast, frame-level labels finely segment expert behavior over time, assigning higher rewards to frames closer to the goal and ensuring more effective progression toward task objectives. Additional analyses, including frame-to-frame alignment between TL and SE, label predictions by $F_{\text{label},f}$ and $F_{\text{label},s}$, and corresponding reward estimates $\hat{R}$, are provided in Appendix E. The appendix further reports trajectory-level analyses for other environments, which consistently validate the Walker-to-Cheetah findings.

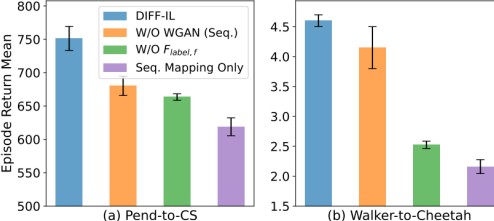

Figure 8: Component evaluation

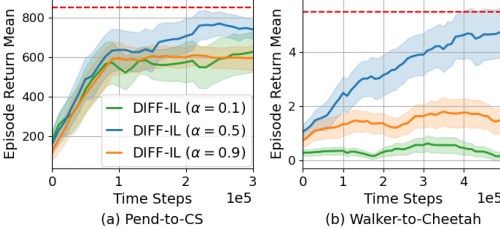

Figure 9: WGAN control factor $\alpha$

### 5.5 ABLATION STUDIES

**Component Evaluation:** To evaluate the components of proposed DIFF-IL, we compare 4 configurations: 'W/O WGAN (Seq.)', excluding sequence-based WGAN losses $\mathcal{L}_{\text{disc},s}$ and $\mathcal{L}_{\text{gen},s}$; 'W/O $F_{\text{label},f}$', omitting frame-wise time labeling while retaining per-frame feature extraction; 'Seq. Mapping Only', using only sequence-based mapping and labeling; and 'DIFF-IL', the full method. Fig. 8 compares performance in Walker-to-Cheetah and Pend-to-CS, where DIFF-IL shows the superior performance. Result shows that 'Seq. Mapping Only' fails to adapt effectively, while 'W/O WGAN (Seq.)' and 'W/O $F_{\text{label},f}$' show moderate improvements. DIFF-IL achieves high performance compared to other setups by combining per-frame domain-invariant feature extraction and frame-wise time labeling to favor later frames, showing their impact on domain adaptation and task success.

**WGAN Control Factor $\alpha$:** To investigate the impact of hyperparameters in DIFF-IL, we conducted an ablation study on WGAN-related hyperparameters. Here, we examine the WGAN control factor $\alpha$, which balances per-frame and sequence-level mapping in DIFF-IL. Fig. 9 compares performance in Walker-to-Cheetah and Pend-to-CS for $\alpha = 0.1$, $0.5$, and $0.9$. The results indicate that $\alpha = 0.5$ achieves the best performance, validating it as the default setting. Lower or higher values reduce performance, highlighting the need for balanced per-frame and sequence-level domain adaptation for effective feature extraction. Additional ablation studies for key hyperparameters of DIFF-IL across more environments are provided in Appendix F.

## 6 LIMITATION

Although the proposed DIFF-IL demonstrates strong imitation learning performance, it still has some limitations. First, DIFF-IL introduces several training coefficients and alternates between representation learning and policy optimization. Achieving a good balance between model updates and RL updates is important, but this is a necessary design choice to stabilize learning and can be addressed with reasonable tuning. Second, the method learns separate frame and sequence embedding networks from observations, which increases training time. However, this design is critical for exact distribution matching, and the performance improvements over baselines are substantial. Moreover, compared with a closely related algorithm such as D3IL, the overall wall-clock training time is much faster in practice, indicating that the added complexity does not pose a significant practical limitation.

## 7 CONCLUSION

In this paper, we proposed DIFF-IL, a novel cross-domain IL framework that effectively addresses the challenges of image-based observation. By combining per-frame feature extraction with frame-wise time labeling, DIFF-IL removes domain-specific artifacts while preserving task-relevant features, enabling robust alignment and successful policy transfer even under significant domain gaps. Extensive experiments demonstrate that DIFF-IL achieves superior domain-invariant representation learning and accurate behavior imitation across diverse settings, including challenging robot manipulation tasks with embodiment changes and resolution-shifted inputs that simulate real deployment conditions. These results establish DIFF-IL as a strong foundation for advancing cross-domain IL in visually demanding applications.

## ETHICS STATEMENT

This work develops DIFF-IL, a cross-domain imitation learning framework evaluated entirely within controlled simulated environments (MuJoCo and DMC). The study does not involve real-world system interaction, human participants, or personally identifiable data, thereby avoiding risks related to safety or privacy. Moreover, DIFF-IL focuses on learning domain-invariant representations rather than memorizing domain-specific patterns, which helps mitigate potential distributional biases. Given its exclusive reliance on simulation data and its goal of advancing methodology rather than direct deployment, we do not identify any negative ethical concerns associated with this research.

## REPRODUCIBILITY STATEMENT

We made significant efforts to ensure the reproducibility of our results. Section 5 provides a detailed description of the proposed DIFF-IL framework. Appendices B and C contain comprehensive implementation details including network architectures, hyperparameters, and training procedures. All benchmarks used in the experiments are publicly available and well-documented in the literature. An anonymized code repository with complete implementation and experimental scripts has been submitted as supplementary material. Baseline methods and their official implementations are detailed in Appendix C.2, allowing for independent reproduction of all results reported in this paper.

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

## A    THE USE OF LARGE LANGUAGE MODELS

In this study, LLMs were used exclusively for polishing and improving the readability of the manuscript. Their role was strictly limited to addressing typographical and grammatical issues, and they were not involved in generating research ideas, designing experiments, analyzing results, or developing the core arguments of the work.

## B    DETAILED IMPLEMENTATION AND ALGORITHM OF DIFF-IL

In this section, we detail the implementation of the proposed methods in DIFF-IL. Section B.1 redefines the loss functions, incorporating loss scales and network parameters. Section B.2 provides the implementation details of the Gradient Penalty (GP) used in WGAN. Section B.3 explains the implementation of RL losses for training the target learner policy. Section B.4 details the specific architectures of the networks introduced in Section B.1, including their structural design and parameter configurations. Finally, Section B.5 describes the algorithm of the proposed DIFF-IL framework.

### B.1    REDEFINED LOSS FUNCTIONS FOR DIFF-IL

In this section, we redefine the losses in DIFF-IL, explicitly including their associated parameters, as described in Section 4. The encoder is parameterized as $\phi$, the domain-specific decoders as $\psi^S$ (source) and $\psi^T$ (target), the frame and sequence discriminators as $\zeta_f$ and $\zeta_s$, and the frame and sequence label discriminators as $\chi_f$ and $\chi_s$, respectively.

The unified WGAN losses for the discriminator and generator are redefined as:

$$
\mathcal{L}_{\text{disc}}(\zeta_f, \zeta_s) := \lambda_{\text{disc}} \cdot \mathbb{E}\left[\alpha\underbrace{(-D_{\zeta_f}(z_t^S) + D_{\zeta_f}(z_t^T))}_{L_{\text{disc},f}(\zeta_f)} + (1-\alpha)\underbrace{(-D_{\zeta_s}(z_{\text{seq},t}^S) + D_{\zeta_s}(z_{\text{seq},t}^T))}_{L_{\text{disc},s}(\zeta_s)}\right] + \lambda_{\text{gp}} \cdot GP
$$

(B.1)

$$
\mathcal{L}_{\text{gen}}(\phi) := \lambda_{\text{gen}} \cdot \mathbb{E}\left[\alpha\underbrace{(D_{\zeta_f}(z_t^S) - D_{\zeta_f}(z_t^T))}_{L_{\text{gen},f}(\phi)} + (1-\alpha)\underbrace{(D_{\zeta_s}(z_{\text{seq},t}^S) - D_{\zeta_s}(z_{\text{seq},t}^T))}_{L_{\text{gen},s}(\phi)}\right]
$$

(B.2)

where $GP$ is gradient penalty term, $z_t^d \sim p_\phi(\cdot|o_t^d)$ and $z_{\text{seq},t}^d \sim p_\phi(\cdot|o_{\text{seq},t}^d)$ for $d \in \{S,T\}$. The coefficients $\lambda_{\text{disc}}$, $\lambda_{\text{gen}}$, and $\lambda_{\text{gp}}$ control the contributions of the losses, while $\alpha$ balances frame- and sequence-based mappings.

The encoder-decoder loss, incorporating generator, reconstruction, and feature consistency losses, is redefined as:

$$
\mathcal{L}_{\text{enc-dec}}(\phi, \psi^S, \psi^T) := \sum_{d=S,T} \mathbb{E}_{z_t^d \sim p_\phi(\cdot|o_t^d)}\left[\lambda_{\text{recon}} \cdot \underbrace{\|o_t^d - \hat{o}_t^d\|_2}_{\text{Reconstruction Loss}} + \lambda_{\text{fcon}} \cdot \underbrace{\|\bar{z}_t - \hat{z}_t^d\|_2}_{\text{Feature Consistency Loss}}\right],
$$

(B.3)

where $d'$ is the opposite domain of $d$, $\hat{o}_t^d = \psi^d(z_t^d)$ and $\hat{z}_t^d \sim p_\phi(\cdot|\psi^{d'}(z_t^d))$, with coefficients $\lambda_{\text{recon}}$ and $\lambda_{\text{fcon}}$ controlling reconstruction and feature consistency losses. The sequence label loss and the frame-wise time labeling loss are redefined as:

$$
\mathcal{L}_{\text{label},s}(\phi, \chi_s) := \sum_{d=S,T} \lambda_{\text{label},s}^d \cdot \mathbb{E}_{z_{\text{seq},t}^d \sim p_\phi(\cdot|o_{\text{seq},t}^d)}\left[\text{BCE}(\mathbb{1}_{o_{\text{seq},t}^d \sim \mathcal{B}^{SE}}, F_{\chi_s}(z_{\text{seq},t}^d))\right],
$$

(B.4)

$$
\mathcal{L}_{\text{label},f}(\chi_f) := \lambda_{\text{label},f} \cdot \mathbb{E}_{z_t^S \sim p_\phi(\cdot|o_t^S)}\left[\text{BCE}(y_t, F_{\chi_f}(z_t^S))\right],
$$

(B.5)

where $\lambda_{\text{label},s}^d$ is the sequence label loss coefficient and $y_t$ is the time label for frame $o_t^S$. Finally, the reward is redefined as:

$$
\hat{R}_t = -\log(1 - F_{\chi_s}(z_{\text{seq},t+1}^T) \cdot F_{\chi_f}(z_t^T) + \epsilon)
$$

(B.6)

where $\epsilon = 1 \times 10^{-12}$ prevents numerical issues when the product of the sequence and frame labels approaches 1. Details of the loss scale coefficients for all losses are summarized in Appendix C.4.

### B.2 IMPLEMENTATION OF GP

To ensure stable training of the adversarial network, the WGAN framework Gulrajani et al. (2017a) incorporates a gradient penalty (GP) to enforce 1-Lipschitz continuity for the discriminator. In the redefined discriminator loss in Eq. equation B.1, the GP term can be defined as follows:

$$\text{Gradient Penalty} = \left( \left\| \alpha \cdot \nabla_{\delta_{\text{label},f}} D_{\zeta_f}(\delta_{\text{label},f}) + (1-\alpha) \cdot \nabla_{\delta_{\text{label},s}} D_{\zeta_s}(\delta_{\text{label},s}) \right\|_2 - 1 \right)^2, \quad \text{(B.7)}$$

where $\delta_{\text{label},f}$ and $\delta_{\text{label},s}$ are the interpolated features between the source and target domain features, computed as:

$$\delta_{\text{label},f} = \delta z_t^S + (1-\delta)z_t^T, \quad \text{(B.8)}$$
$$\delta_{\text{label},s} = \delta z_{\text{seq},t}^S + (1-\delta)z_{\text{seq},t}^T. \quad \text{(B.9)}$$

Here, the frame features $z_t^S \sim p_\phi(\cdot|o_t^S)$ and $z_t^T \sim p_\phi(\cdot|o_t^T)$, and the sequence features $z_{\text{seq},t}^S \sim p_\phi(\cdot|o_{\text{seq},t}^S)$ and $z_{\text{seq},t}^T \sim p_\phi(\cdot|o_{\text{seq},t}^T)$, represent features extracted from the source and target domains, respectively. The scalar $\delta \sim \text{Unif}(0,1)$ serves as the interpolation factor. The GP term enforces Lipschitz continuity on the discriminator, stabilizing adversarial training by mitigating extreme gradients and promoting smooth convergence. Additionally, we maintain a 5:1 training ratio between the discriminator and generator, following standard practices to ensure stability during training.

### B.3 RL IMPLEMENTATION

To train the target learning policy $\pi^{TL}$, we parameterize both the policy $\pi^{TL}$ and the state-action value function $Q$ using parameter $\theta$. Utilizing Soft Actor-Critic (SAC) Haarnoja et al. (2018), the critic and actor losses are defined as follows:

$$\mathcal{L}_Q(\theta) = \mathbb{E}_{(s_t,a_t,s_{t+1},o_{seq,t}^T) \sim \mathcal{B}^{TL}} \left[ \frac{1}{2} \Big( Q_\theta(s_t,a_t) - \Big( \hat{R}_t + \gamma \, \mathbb{E}_{a_{t+1} \sim \pi_\phi(\cdot|s_{t+1})} \Big[ \right.$$
$$\left. Q_{\theta^-}(s_{t+1},a_{t+1}) - \lambda_{\text{ent}} \cdot \log \pi_\theta(\cdot|s) \Big] \Big) \Big)^2 \right], \quad \text{(B.10)}$$

$$\mathcal{L}_\pi(\theta) = \mathbb{E}_{s_t \sim \mathcal{B}^{TL}} \left[ \mathbb{E}_{a_t \sim \pi_\theta(\cdot|s_t)} \left[ D_{KL}\Big( \pi_\theta(a_t|s_t) || \frac{\exp(Q_\theta(s_t,a_t)/\lambda_{\text{ent}})}{Z_\theta(s_t)} \Big) \right] \right], \quad \text{(B.11)}$$

where $D_{KL}$ represents the Kullback-Leibler (KL) divergence, $Q_\theta(s,a)$ denotes the parameterized state-action value function, $\theta^-$ is the parameter of the target network updated via the exponential moving average (EMA) method, $\pi_\theta(a|s)$ represents the target learner policy parameterized by $\theta$, and $\hat{R}_t$ is computed as in Eq. B.6, capturing the estimated effect of actions. The critic loss minimizes the difference between the predicted value $Q_\theta$ and the target value derived from the Soft Bellman equation, ensuring accurate value estimation. The actor loss minimizes the divergence between the policy $\pi_\theta$ and the Softmax distribution induced by $Q$, encouraging the policy to prioritize actions that maximize long-term rewards. To enhance training stability, SAC incorporates double Q-learning and automatic adjustment of the entropy coefficient $\lambda_{\text{ent}}$.

## B.4    NETWORK ARCHITECTURE AND CONFIGURATIONS

This subsection outlines the architecture of the networks used in DIFF-IL, detailing the encoder, decoder, discriminators, label networks, and the SAC-based actor-critic structure, as follows:

- **Encoder** ($p_\phi$): A convolutional neural network that extracts features from input data. It comprises convolutional layers with 16, 32, and 64 filters, applied with different strides, and utilizes LeakyReLU activations Xu (2015) to mitigate vanishing gradient issues. The final output is flattened and passed through a dense layer with 32 units.

- **Decoders** ($q_{\psi^S}^S, q_{\psi^T}^T$): Reconstructs data from encoded features using `ConvTranspose` (transposed convolutional layers) with 64 and 32 filters. It upsamples feature maps back to their original resolution, ending with a `ConvTranspose` layer outputting a 3-channel image. The final layer uses a linear activation for reconstruction.

- **WGAN discriminators** ($D_{\zeta_f}, D_{\zeta_s}$): These discriminators distinguish between source and target features from the encoder, operating on either frame or sequence level. They consist of dense layers with LeakyReLU activations and a final dense layer without activation, producing a scalar output indicating whether the input features are from the source or target domain.

- **Label discriminators** ($F_{\chi_f}, F_{\chi_s}$): Predict labels for frames and sequences using dense layers with LeakyReLU activations. The final layer applies a sigmoid activation to output probabilities for the class labels.

- **Critic** ($Q_\theta$): Evaluates the value of actions using dense layers with ReLU activations. The critic outputs the state-action value for each action.

- **Target learner policy** ($\pi_\theta$): Generates actions modeled as independent Gaussian distributions for each action dimension. The policy network outputs the mean $\mu_\theta$ and standard deviation $\sigma_\theta$, both the mean and standard deviation have sizes equal to the action dimension. This stochastic formulation enables action sampling, facilitating exploration during training.

Details about the action dimensions for each environment are available in Appendix C.3, and a summary of the network architecture is presented in Table B.1.

| Network | Layers | Network | Layers |
|---|---|---|---|
| **Encoder** ($p_\phi$) | Conv(16, 1, LeakyReLU)
Conv(16, 2, LeakyReLU)
Conv(32, 1, LeakyReLU)
Conv(32, 2, LeakyReLU)
Conv(64, 1, LeakyReLU)
Conv(64, 2, LeakyReLU)
Flatten
Dense(32) | **Decoders** ($q_{\psi^S}^S, q_{\psi^T}^T$) | ConvTranspose(64, 1, LeakyReLU)
ConvTranspose(64, 2, LeakyReLU)
ConvTranspose(32, 1, LeakyReLU)
ConvTranspose(32, 2, LeakyReLU)
ConvTranspose(16, 1, LeakyReLU)
ConvTranspose(16, 2, LeakyReLU)
ConvTranspose(3, 1) |
| **WGAN discriminators** ($D_{\zeta_f}, D_{\zeta_s}$) | BatchNorm()
Dense(400, LeakyReLU)
Dense(300, LeakyReLU)
Dense(1) | **Label discriminators** ($F_{\chi_f}, F_{\chi_s}$) | BatchNorm()
Dense(400, LeakyReLU)
Dense(300, LeakyReLU)
Flatten
Dense(1, Sigmoid) |
| **Critic** ($Q_\theta$) | Dense(256, ReLU)
Dense(256, ReLU)
Dense(1) | **Target learner policy** ($\pi_\theta$) | Dense(256, ReLU)
Dense(256, ReLU)
Dense(2×Action Dim.) |

Table B.1: Architectural specifications of the proposed networks. Conv(nc, stride, act) represents a convolutional layer with nc filters, stride, and activation act. ConvTranspose(nc, stride, act) denotes a transposed convolutional layer. Flatten reshapes the input into a 1D vector. Dense(nc) indicates a dense layer with nc filters.

## B.5 DIFF-IL ALGORITHM

---

**Algorithm 1** DIFF-IL Framework

---

**Input:** Source domain data $\mathcal{B}^S$, Target domain data $\mathcal{B}^T$
Initialize $p, q^S, q^T, D_f, D_s, F_{\text{label},f}, F_{\text{label},s}, \pi^{TL}$
**for** Iteration $i = 1$ **to** $N_{\text{iter}}$ **do**
    **for** Model training step $k = 1$ **to** $N_{\text{model,train}}$ **do**
        Sample $(o_{\text{seq},t}^S, o_{\text{seq},t}^T) \sim (\mathcal{B}^S, \mathcal{B}^T)$
        Calculate $\mathcal{L}_{\text{disc},f}$ and $\mathcal{L}_{\text{disc},s}$
        Update $D_f$ and $D_s$ using $\mathcal{L}_{\text{disc},f}$ and $\mathcal{L}_{\text{disc},s}$
        **if** $k \mod n = 0$ **then**
            Calculate $\mathcal{L}_{\text{enc-dec}}, \mathcal{L}_{\text{gen},f}, \mathcal{L}_{\text{gen},s}, \mathcal{L}_{\text{label},f}, \mathcal{L}_{\text{label},s}$
            Update $p, q^S, q^T, F_{\text{label},f}$, and $F_{\text{label},s}$ based on the calculated loss functions
        **end if**
    **end for**
    **for** RL training step $l = 1$ **to** $N_{\text{RL,train}}$ **do**
        Compute reward $\hat{R}_t$ using $F_{\text{label},f}$ and $F_{\text{label},s}$
        Perform RL and update the target learner $\pi^{TL}$
    **end for**
    Store transitions generated by $\pi^{TL}$ in $\mathcal{B}^{TL}$
**end for**

---

## C DETAILED EXPERIMENTAL SETUP

This section provides the necessary details for conducting the experiments. Section C.1 outlines the experimental setup and the design of the ablation study to analyze the impact of key hyperparameters. Section C.2 provides a brief overview of the baseline IL algorithms used for performance comparison. Section C.3 details the environments used in the experiments. Finally, Section C.4 explains the hyperparameters of DIFF-IL and summarizes the optimal configurations.

### C.1 EXPERIMENTAL SETUP

Prior to training, we construct datasets essential for the learning process. Buffer sizes for $\mathcal{B}^{SE}, \mathcal{B}^{SR}, \mathcal{B}^{TL}$, and $\mathcal{B}^{TR}$ are fixed at 50K. Among these, $\mathcal{B}^{SE}, \mathcal{B}^{SR}$, and $\mathcal{B}^{TR}$ remain static during training, while $\mathcal{B}^{TL}$ is dynamically updated. After each model and RL training epoch, $\mathcal{B}^{TL}$ is refreshed with 1,000 new samples from environment interactions, replacing the oldest data. Initially, $\mathcal{B}^{TL}$ is populated with random samples, similar to $\mathcal{B}^{TR}$. To construct $\mathcal{B}^{SE}$, we train the source expert policy $\pi^{SE}$ using SAC Haarnoja et al. (2018) and collect samples from $\pi^{SE}$. For $\mathcal{B}^{SR}$ and $\mathcal{B}^{TR}$, random policies are used for data collection. In tasks like IP, IDP, Pendulum, CS, and Acrobot, where random policies can sustain extended downward pole positions, episode lengths vary between expert and random policies. Detailed specifications are provided in Section C.3.

The implementation is based on TensorFlow 2.5 with CUDA 11.4 and CUDNN 8.2.4, running on an AMD EPYC 7313 CPU with an NVIDIA GeForce RTX 3090 GPU. GPU memory usage is approximately 9GB for Pendulum tasks and 18GB for Mujoco tasks, influenced by batch size and feature dimensions. Each epoch requires about one minute. The codebase builds on DeGAIL Cetin & Celiktutan (2021): `https://github.com/Aladoro/domain-robust-visual-il`.

### C.2 OTHER CROSS-DOMAIN IL METHODS

In this section, we briefly describe the approaches of the four cross-domain algorithms compared with our method:

- **TPIL** (Stadie et al., 2017) addresses domain shift in imitation learning by combining unsupervised domain adaptation (Ganin & Lempitsky, 2015) with GAIL (Ho & Ermon, 2016). It uses an encoder to extract domain-independent features, a domain discriminator to differentiate domains, and a label discriminator to classify expert and non-expert behaviors. A gradient reversal layer optimizes these components simultaneously, aligning features across domains for effective policy learning. Code: `https://github.com/bstadie/third_person_im`.
- **DeGAIL** Cetin & Celiktutan (2021) extracts domain-free features by reducing mutual information between source and target domain data passed through the same encoder. It trains the encoder to minimize domain-related information while using GAIL for reward estimation and reinforcement learning. Code: `https://github.com/Aladoro/domain-robust-visual-il`.
- **GWIL** Fickinger et al. (2022) leverages the Gromov-Wasserstein distance Mémoli (2011) as a direct reward, learning optimal coupling between expert and imitator state-action spaces. This distance measures action similarity and guides policy optimization via policy gradient methods. Code: `https://github.com/facebookresearch/gwil`.
- **D3IL** Choi et al. (2024) enhances feature extraction using dual encoders for domain-specific and behavior-specific features, with discriminators refining extraction accuracy through cycle-consistency and reconstruction. A discriminator generates rewards by distinguishing between expert and learner behaviors. Code: `https://github.com/sunghochoi122/D3IL`.

For the RL implementation of the target learner policy, all cross-domain IL algorithms are implemented using SAC Haarnoja et al. (2018). Although TPIL originally employs TRPO Schulman et al. (2017), we re-implemented it using SAC to ensure a fair comparison, following the approach suggested in the D3IL paper Choi et al. (2024).

### C.3 ENVIRONMENTAL SETUP

This section outlines the experimental environments categorized into Pendulum Tasks, MuJoCo Tasks and Robot Manipulation Tasks. Pendulum Tasks include the Inverted Pendulum (IP) and Inverted

Double Pendulum (IDP) from the MuJoCo 150 library, along with modified MuJoCo Reacher environments. Reacher2 (RE2) modifies the goal point of the Reacher environment, while Reacher3 (RE3) extends the arm joint configuration to three joints. Additional environments from the DeepMind Control Suite (DMC) include Pendulum, Cartpole Swingup (CS), and Acrobot. MuJoCo Tasks are adapted DMC environments with a fixed distant camera viewpoint for image-based observations and redesigned reward functions focusing solely on agent velocity. Fig. C.1 shows image observations for Pendulum environments, Fig. C.2 provides those for MuJoCo environments, and Fig. C.3 illustrates image observations for Robot Manipulation tasks, including both robot arm and humanoid-like arm scenarios including low-resolution to high-resolution tasks.

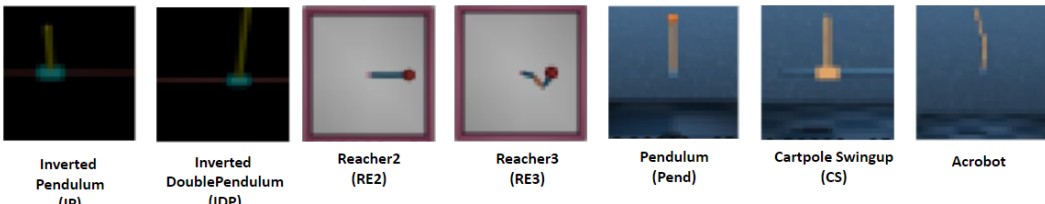

Figure C.1: Image observation of Pendulum environments

**Pendulum Tasks**

- **Inverted Pendulum (IP)**: The Inverted Pendulum task requires balancing a single pole in an upright position. The agent controls the pole's angle, angular velocity, and the cart's position and velocity. The state space $\mathcal{S}$ is 4-dimensional, while the action space $\mathcal{A}$ is 1-dimensional, representing the force applied to the cart. Rewards increase as the pole remains closer to vertical. Observations are $32 \times 32$ RGB images. Random episodes are $H_{\tilde{\tau}} = 50$, and expert/learner episodes are $H_{\tilde{\tau}} = 1000$.

- **Inverted Double Pendulum (IDP)**: The IDP extends the IP task to two interconnected poles. The state space $\mathcal{S}$ is 11-dimensional, including angles and angular velocities of both poles and the cart's position and velocity. The action space $\mathcal{A}$ remains 1-dimensional. Rewards increase when both poles are upright. Observations are $32 \times 32$ RGB images. Random episodes are $H_{\tilde{\tau}} = 50$, and expert/learner episodes are $H_{\tilde{\tau}} = 1000$.

- **Reacher Tasks (RE2, RE3)**: These tasks involve controlling a robotic arm with two (RE2) or three (RE3) joints to reach one of 16 randomly assigned targets. The target position is defined in polar coordinates, with $r \in 0.15, 0.2$ and $\varphi \in 0, \pi/4, \pi/2, \ldots, 7\pi/4$. The state space $\mathcal{S}$ is 11-dimensional for RE2 and 14-dimensional for RE3, while the action spaces $\mathcal{A}$ have 2 and 3 dimensions, respectively. Negative rewards reflect the distance between the end effector and the target, with zero awarded for reaching the target. Observations are $48 \times 48$ RGB images. For all scenarios, $H_{\tilde{\tau}} = 50$ for random, expert, and learner episodes.

- **Pendulum (Pend)**: The Pendulum task involves balancing a single pole attached to a fixed pivot point. The state space $\mathcal{S}$ is 3-dimensional, and the action space $\mathcal{A}$ is 1-dimensional, representing the torque applied to the pivot. Rewards increase as the pole remains upright. Observations are $32 \times 32$ RGB images. Random episodes are $H_{\tilde{\tau}} = 200$, and expert/learner episodes are $H_{\tilde{\tau}} = 1000$.

- **Cartpole Swingup (CS)**: The CS task requires balancing a pole on a cart moving along a horizontal axis. The state space $\mathcal{S}$ is 5-dimensional, and the action space $\mathcal{A}$ is 1-dimensional, representing the force applied to the cart. Rewards increase when the pole stays upright. Observations are $32 \times 32$ RGB images. Random episodes are $H_{\tilde{\tau}} = 200$, and expert/learner episodes are $H_{\tilde{\tau}} = 1000$.

- **Acrobot**: The Acrobot task involves controlling two connected poles to achieve an upright position from a random position start. The state space $\mathcal{S}$ is 6-dimensional, and the action space $\mathcal{A}$ is 1-dimensional, representing the torque applied to the joint connecting the poles. Rewards increase when the poles reach vertical alignment. Observations are $32 \times 32$ RGB images. Random episodes are $H_{\tilde{\tau}} = 200$, and expert/learner episodes are $H_{\tilde{\tau}} = 1000$.

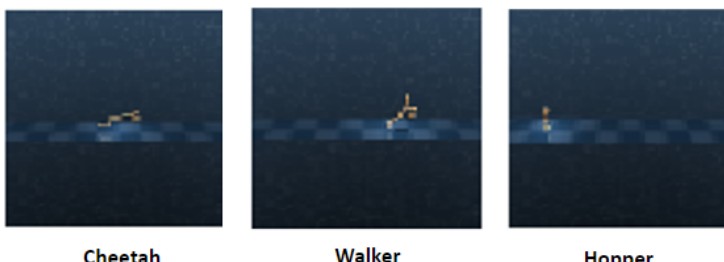

Figure C.2: Image observation of MuJoCo environments

**MuJoCo Tasks**

- **Cheetah**: The Cheetah environment features a quadrupedal agent designed for fast and efficient running. The state space $\mathcal{S}$ is 17-dimensional, encoding joint angles, velocities, and torso orientation, while the action space $\mathcal{A}$ is 6-dimensional, representing torques applied to joints. Observations are $64 \times 64$ RGB images captured from a fixed camera. The reward function depends solely on forward velocity, aligning with the task's objective. Random and expert/learner episodes are $H_{\tilde{\tau}} = 200$.

- **Walker**: The Walker environment involves a bipedal agent simulating human-like locomotion. Its state space $\mathcal{S}$ is 24-dimensional, including joint angles, velocities, and torso orientation. The action space $\mathcal{A}$ has 6 dimensions, controlling joint torques. Observations are $64 \times 64$ RGB images from a fixed camera. The reward function is modified to depend only on forward velocity. Random and expert/learner episodes are $H_{\tilde{\tau}} = 200$.

- **Hopper**: The Hopper environment tasks a single-legged agent with moving forward efficiently. The state space $\mathcal{S}$ is 15-dimensional, capturing joint positions, velocities, and torso orientation. The action space $\mathcal{A}$ is 4-dimensional, representing joint torques. Observations are $64 \times 64$ RGB images taken from a fixed camera. The reward function relies exclusively on forward velocity, emphasizing efficient locomotion. Random and expert/learner episodes are $H_{\tilde{\tau}} = 200$.

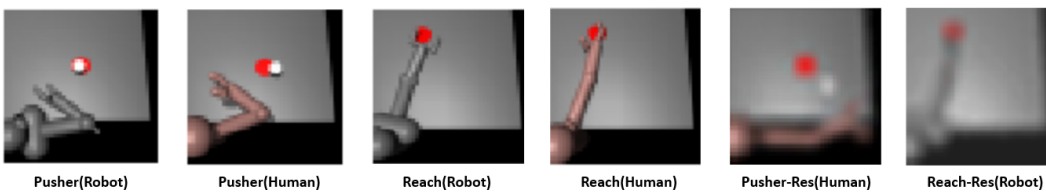

Figure C.3: Image observation of Robot Manipulation environments

**Robot Manipulation Tasks**

- **Pusher**: The Pusher environment features a 7-DoF robotic arm that pushes an object toward a goal. Its state space $\mathcal{S}$ is 27-dimensional, including joint angles and velocities, as well as the 3D positions of the fingertip, object, and goal. The action space $\mathcal{A}$ is 7-dimensional, corresponding to continuous torques at each joint. Observations are $48 \times 48$ RGB images from a fixed viewpoint. The reward function is modified to depend solely on the distance between the object and the goal. Each episode consists of $H_{\tilde{\tau}} = 200$ time steps. The Humanoid-like variant adds a fingertip without joints, affecting only physical interactions while leaving the state and action dimensions unchanged. The suffix **R** denotes a robot task and **H** denotes a humanoid-like task (e.g., Pusher:R, Pusher:H).

- **Reach**: The Reach environment is configured identically to Pusher, except that the agent moves its robotic arm's fingertip as close as possible to a designated goal point without manipulating any object. The reward is based solely on the distance between the fingertip and the goal.

- **Resolution Shifts**: Resolution Shifts are not treated as independent tasks but as an additional setup to examine robustness under varying image resolutions that may occur in real-world scenarios. To construct this setup without extra data collection, To construct this setup without extra data collection, we first downsample source-domain images to $(32, 32)$ using bicubic interpolation and then upsample them back to the original resolution using the same bicubic method. Although this

procedure can be extended to other environments, we focus on the high-DoF Robot Manipulation suite and evaluate Resolution Shifts across all its tasks to emphasize practical relevance for real-world deployment. In this context, the '-Res' suffix appended to each task name denotes the setting where source domain images are downsampled and reconstructed to lower-resolution inputs.

The state dimensions, action dimensions, image sizes, and episode lengths for all environments are summarized in Table C.1. Image resolution for each task was configured to the minimum level required for clear agent distinction, optimizing memory usage while maintaining sufficient visual detail.

| Environment | $\mathcal{S}$ dim. | $\mathcal{A}$ dim. | Image size | Expert Epi. length ($H_{\hat{\tau}}$) | Random Epi. length |
|---|---|---|---|---|---|
| IP | 4 | 1 | 32x32 | 1000 | 50 |
| IDP | 11 | 1 | 32x32 | 1000 | 50 |
| RE2 | 11 | 2 | 48x48 | 50 | 50 |
| RE3 | 14 | 3 | 48x48 | 50 | 50 |
| Pend | 3 | 1 | 32x32 | 1000 | 200 |
| CS | 5 | 1 | 32x32 | 1000 | 200 |
| Acrobot | 6 | 1 | 32x32 | 1000 | 200 |
| Cheetah | 17 | 6 | 64x64 | 200 | 200 |
| Walker | 24 | 6 | 64x64 | 200 | 200 |
| Hopper | 15 | 4 | 64x64 | 200 | 200 |
| Pusher:R | 27 | 7 | 48x48 | 200 | 200 |
| Pusher:H | 27 | 7 | 48x48 | 200 | 200 |
| Reach:R | 27 | 7 | 48x48 | 200 | 200 |
| Reach:H | 27 | 7 | 48x48 | 200 | 200 |

Table C.1: State, action, and image sizes used in the experiments section. Resolution-shift settings also use the same image size because low-resolution inputs are upsampled prior to evaluation.

## C.4 HYPERPARAMETER SETTINGS

In this subsection, we address the hyperparameters used in the implementation. These hyperparameters are summarized in Tables C.2, C.3 and C.4, C.5. Environment-specific state, action, and image dimensions are in Table C.1. Task names are abbreviated in the tables for clarity: in Pendulum tasks, 'Pend' is 'P', and 'Acrobot' is 'A'; in MuJoCo tasks, 'Cheetah' is 'C', 'Walker' is 'W', 'Hopper' is 'H'; in Robot Manipulation tasks, suffix 'R' is a robot, suffix 'H' is humanoid-like, with '-' omitted. For Robot Manipulation tasks, the resolution-shifted variants (denoted with '-Res' suffix) use identical hyperparameters to their standard counterparts, differing only in source domain image resolution. As described in the main text, we conducted hyperparameter sweeps only for WGAN-related parameters. All other hyperparameters were fixed at appropriate values, as provided in the tables.

For WGAN-related losses, the discriminator loss weight was searched in the range of 0.01 to 50, while the generator loss weight was searched from 0.01 to 10, with the best hyperparameters selected for each task. Although the search range may seem broad, adjustments were made based on feature mapping quality: when features from the domains did not overlap sufficiently, the discriminator loss was reduced or the generator loss was increased; conversely, when excessive mapping caused the target learner's features to overly align with the expert's, the discriminator loss was increased or the generator loss was reduced. These adjustments are discussed in greater detail in the ablation study in Appendix F.

Also, the WGAN control coefficient $\alpha$, which balances the ratio of frame and sequence mapping, consistently performed best at 0.5 across all tasks and was fixed at this value. For the sequence label discriminator loss scale, the source side was set to a high value of 10, as expert and random data are well separated, while the target side, where label distinctions are less clear, was set to a much smaller value of $1e-3$.

| shard hyperparameter / Task | Pendulum tasks | MuJoCo tasks | Robot tasks |
|---|---|---|---|
| Reconstruction ($\lambda_{\text{recon}}$) | 0.5 | 1 (HtoW 0.5) | 1 |
| Feature Consistency ($\lambda_{\text{fcon}}$) | 1 (IPtoIDP, IDPtoIP 0.1) | 1 | 1 |
| Gradient Penalty ($\lambda_{\text{GP}}$) | 10 | 10 | 10 |
| WGAN control coefficient ($\alpha$) | 0.5 | 0.5 | 0.5 |
| Sequence label Discriminator (Source, $\lambda_{\text{label},s}^{S}$) | 10 | 10 | 10 |
| Sequence label Discriminator (Target, $\lambda_{\text{label},s}^{T}$) | 1e-3 | 1e-3 | 1e-3 |
| Frame Labelnet $\lambda_{\text{label},f}$ | 10 | 10 | 10 |
| Optimizer | 1e-3 | 1e-3 | 1e-3 |

Table C.2: Shared hyperparameters across all tasks. (tasks, value) next to each value indicates exception tasks. Note that "Robot tasks" denotes the Robot Manipulation Tasks.

| Hyperparameter / Task | IPtoIDP | IDPtoIP | RE2toRE3 | RE3toRE2 | PtoCS | PtoA | CStoP | CStoA |
|---|---|---|---|---|---|---|---|---|
| Discriminator ($\lambda_{\text{disc}}$) | 1 | 1 | 1 | 50 | 50 | 1 | 50 | 50 |
| Generator ($\lambda_{\text{gen}}$) | 0.05 | 0.05 | 1 | 1 | 0.5 | 10 | 0.5 | 10 |
| Model batch size | 128 | 128 | 64 | 64 | 128 | 128 | 128 | 128 |
| Model train num | 200 | 200 | 100 | 100 | 100 | 100 | 100 | 100 |
| RL train num | 2000 | 2000 | 2000 | 2000 | 1000 | 1000 | 1000 | 1000 |

Table C.3: Hyperparameter setup for Pendulum tasks

| Hyperparameter / Task | WtoC | CtoW | HtoC | CtoH | WtoH | HtoW |
|---|---|---|---|---|---|---|
| Discriminator ($\lambda_{\text{disc}}$) | 0.5 | 0.02 | 0.1 | 0.05 | 1 | 0.02 |
| Generator ($\lambda_{\text{gen}}$) | 1 | 0.05 | 0.1 | 0.01 | 0.01 | 0.05 |
| Model batch size | 64 | 64 | 64 | 64 | 64 | 64 |
| Model train num | 100 | 100 | 100 | 100 | 50 | 50 |
| RL train num | 1000 | 1000 | 1000 | 1000 | 1000 | 1000 |

Table C.4: Hyperparameter setup for MuJoCo tasks

| Hyperparameter / Task | Pusher:RtoH | Pusher:HtoR | Reach:RtoH | Reach:HtoR |
|---|---|---|---|---|
| Discriminator ($\lambda_{\text{disc}}$) | 1 | 1 | 1 | 1 |
| Generator ($\lambda_{\text{gen}}$) | 0.2 | 0.2 | 0.1 | 0.1 |
| Model batch size | 64 | 64 | 64 | 64 |
| Model train num | 100 | 100 | 100 | 100 |
| RL train num | 1000 | 1000 | 1000 | 1000 |

Table C.5: Hyperparameter setup for Robot Manipulation tasks. Resolution-shifted variants use identical hyperparameters.

# D    ADDITIONAL COMPARATIVE ANALYSIS OF THE PROPOSED D3IL

## D.1    LEARNING CURVES ON PENDULUM TASKS AND ROBOT MANIPULATION TASKS

This section presents the learning curves for Pendulum and MuJoCo Pusher tasks not covered in Sec. 5, with timesteps allocated based on the learning difficulty of each target environment. As shown in Fig. D.1, the proposed DIFF-IL demonstrated strong performance across most of the compared environments. Notably, the proposed method excels in DMC Pendulum tasks, showing a significant advantage in environments such as Pend-to-CS, Pend-to-Acrobot, and CS-to-Pend. It achieves faster convergence and significantly outperforms other cross-domain IL methods. This demonstrates that the proposed algorithm designs rewards more effectively for mimicking expert behavior compared to other IL approaches, aligning with the results presented in the main text. Furthermore, as shown in Fig. D.2, DIFF-IL demonstrates superior performance in Robot Manipulation tasks. Notably, while the Pusher tasks are relatively easy, DIFF-IL achieves strong results in the more challenging Reach tasks for both Robot-to-Human and Human-to-Robot transfer tasks. For the resolution-shifted variants (denoted with '-Res' suffix), DIFF-IL maintains comparable final performance despite increased learning curve oscillations during training. While the learning curves exhibit higher variability due to the added visual complexity from source-target resolution shifts, the final task performance remains

largely unaffected, demonstrating DIFF-IL's robustness to visual degradation challenges commonly encountered in real-world deployment scenarios.

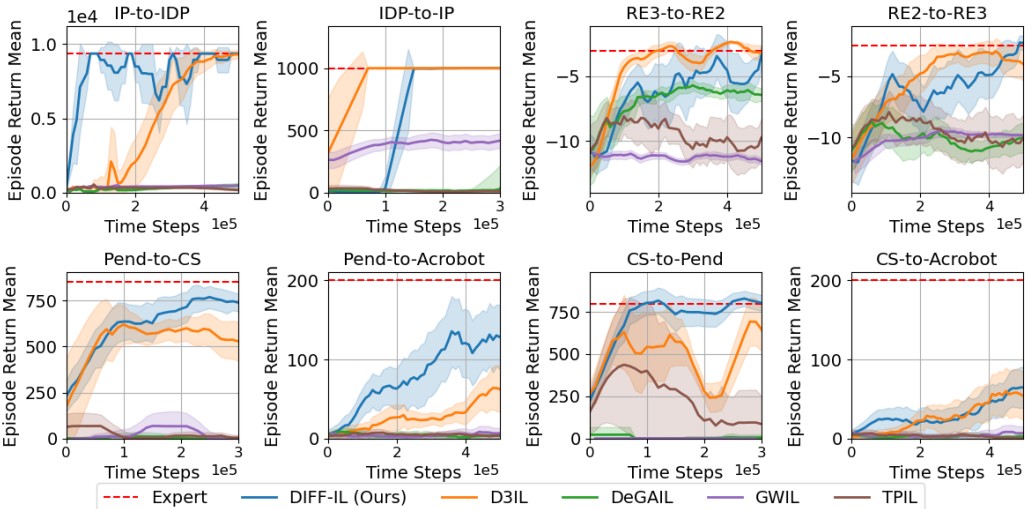

Figure D.1: Performance comparison: Learning curves on Pendulum tasks

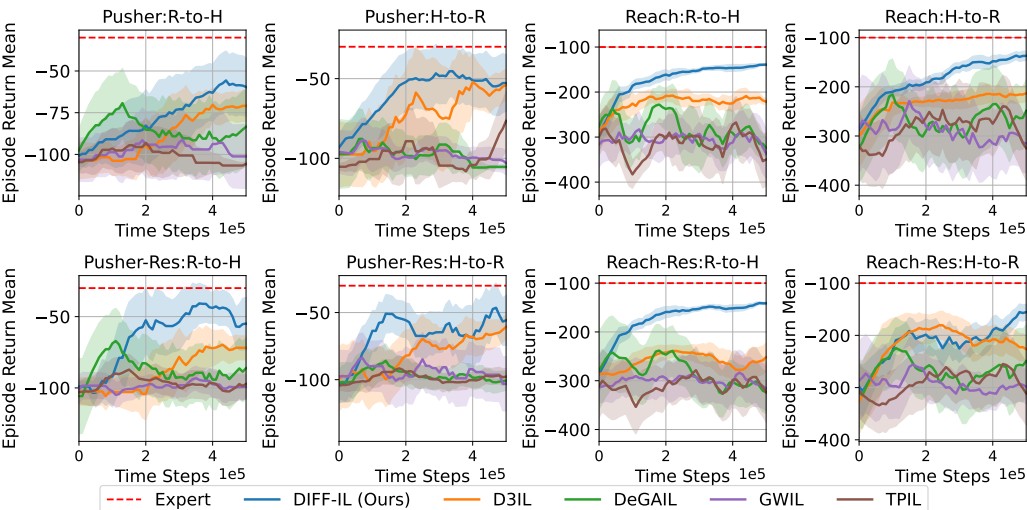

Figure D.2: Performance comparison: Learning curves on Robot Manipulation tasks

## D.2 LIMITED EXPERT DATA

In this section, we evaluate the performance of DIFF-IL under limited access to expert data, reflecting practical challenges in real-world environments. Experiments were conducted on Pend-to-Acrobot and Walker-to-Cheetah tasks, comparing the default setup using 50k images with reduced setups using 5k (10%) and 10k (20%) images. For Pend-to-Acrobot (expert episode length: 1000 steps), 5k and 10k images correspond to 5 and 10 episodes, respectively; for Walker-to-Cheetah (episode length: 200 steps), these correspond to 25 and 50 episodes. All experiments used identical hyperparameter settings to ensure fair evaluation.

| | 5k (10 %) | 10k (20 %) | 50k (default) |
|---|---|---|---|
| Pend-to-Acrobot | $122.33 \pm 40.19$ | $123.61 \pm 41.22$ | $\mathbf{128.24 \pm 40.58}$ |
| Walker-to-Cheetah | $4.40 \pm 1.39$ | $4.47 \pm 1.02$ | $\mathbf{4.54 \pm 0.86}$ |

Table D.1: Ablation study : limited number of expert data

As shown in Table D.1, DIFF-IL maintains strong performance even with much less expert data. Pend-to-Acrobot is largely unaffected by data amount, as its fixed initial states yield similar expert trajectories. Walker-to-Cheetah shows higher variance with less data due to random initial states, but increasing the number of episodes reduces this effect. Nevertheless, no significant performance drop is observed in either task, highlighting DIFF-IL's robustness to limited expert data.

### D.3 Comparing Computational Cost and Training Time

In this subsection, we analyze our computational complexity and training time compared to other baselines. While DIFF-IL introduces additional components such as per-frame encoding and an extra discriminator, it processes the same sequence data as other visual imitation learning (IL) baselines, resulting in only a modest increase in GPU memory usage. For example, in the Pendulum task, TPIL consumed 3GB of memory (with lower performance), DeGAIL and D3IL each required 8GB, while DIFF-IL used 9GB, indicating that its memory overhead remains manageable.

In terms of training time, DIFF-IL achieves a favorable balance between efficiency and performance. TPIL requires 25 seconds per epoch, DeGAIL 45 seconds, D3IL 185 seconds, and DIFF-IL 61 seconds per epoch. While visual observation-based IL methods may be limited by GPU memory requirements in resource-constrained environments, training is typically performed on high-resource servers, and only the final policy is deployed to target devices. Overall, DIFF-IL offers strong performance with reasonable memory and training time requirements.

## E Additional In-depth Analysis for DIFF-IL

This section explains the feature alignment for tasks not covered in Sec. 4. For each task-specific figure, images are aligned by processing the Target Learner (TL) and Source Expert (SE) data through the encoder to extract domain-invariant features. The closest features between SE and TL are then matched for alignment. The images are arranged sequentially from left to right, showing the progression of timesteps. The bottom row displays the changes in Target Random (TR) data over time. Below each image, the estimated frame and sequence label values generated by the frame label discriminator $F_{\text{label},f}$ and sequence label discriminator $F_{\text{label},s}$ are provided. For TL data, the estimated rewards $\hat{R}_t$ calculated using the proposed reward estimation method are also shown. These visualizations highlight how the model achieves alignment and how the discriminators contribute to effective feature mapping and reward assignment throughout the task. Section E.1 discusses cases of training failure, potential mitigation strategies, and future work. Section E.2 presents trajectory analysis for additional tasks not covered in Section 5.

### E.1 Failure Analysis for DIFF-IL

DIFF-IL alternates between a model training step, where domain-invariant features and the label network are learned, and an RL step, where the trained model is used to estimate rewards for target samples and update policy. It is crucial to maintain a balance between acquiring diverse target samples through policy exploration and learning robust feature alignment. If the model is trained too extensively before the policy has collected sufficiently diverse samples, there is a risk that random-like target samples may be aligned with source expert samples, potentially leading to training failure.

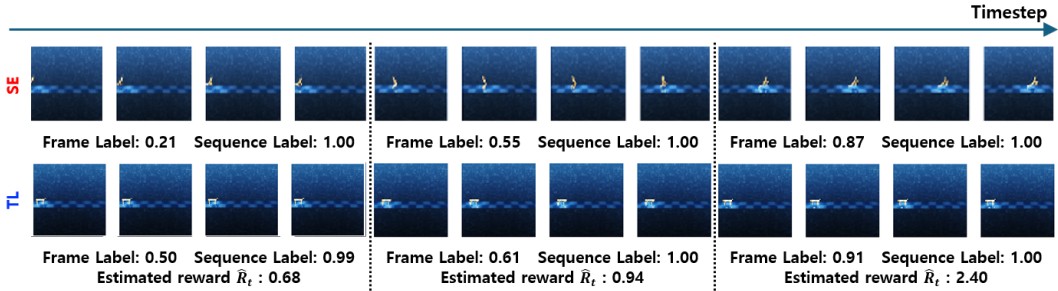

Figure E.1: Failure case feature alignments of Walker-to-Cheetah tasks

Fig. E.1 illustrates a case of trajectory alignment failure in the Walker-to-Cheetah tasks. While the label network successfully learned the expert features, the alignment of target data was inadequate, resulting in samples near the initial position receiving high rewards. Ideally, as the Cheetah agent explores and generates more forward-moving samples, the model should adapt and reflect this progression in its learning. However, if the model is trained excessively or if the policy fails to collect sufficiently diverse samples, the model may learn incorrect alignments. To mitigate this issue, one could reduce the number of model training iterations or incorporate exploration strategies to encourage the policy to acquire more diverse samples. Additionally, leveraging recent advances in generative models to synthesize diverse target samples that better align with the source domain could be a promising direction for future work.

## E.2 ADDITIONAL TRAJECTORY ANALYSES FOR OTHER TASKS

### Pendulum Tasks

- **RE2-to-RE3 task**: Figures E.2 and E.3 illustrate the RE2-to-RE3 task, focusing on image mapping, reward analysis, and label estimation for two scenarios with different goals. In this task, both frame and sequence label values increase as the agent progresses toward the goal, with sequence labels nearing 1 upon reaching the target. Frame label estimates, however, vary based on the agent's proximity to the goal, as the episode continues after the goal is reached. Due to the arm's initial alignment to the right, leftward arm movements are less represented in the expert samples, resulting in higher frame label values for more common rightward movements. Similar to the IP-to-IDP task, random samples that fail to approach the goal produce sequence label values close to 0, enabling effective learning by rewarding the agent for reaching the target quickly and accurately.

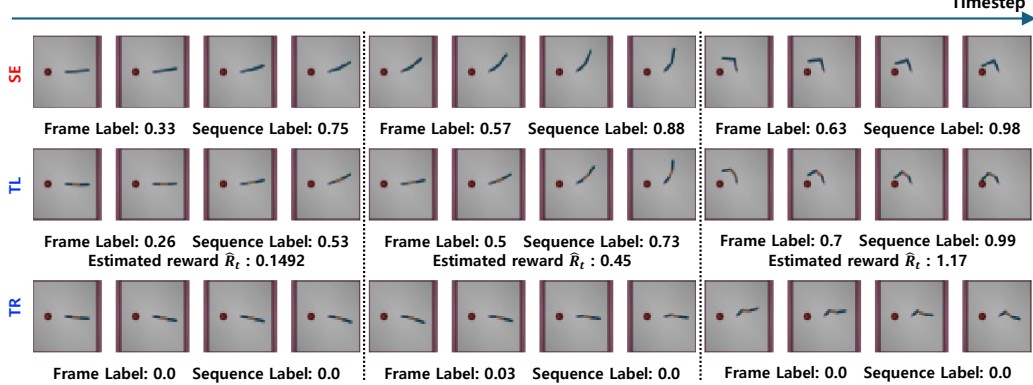

Figure E.2: Image mapping and reward analysis on RE2-to-RE3 task (Goal 1)

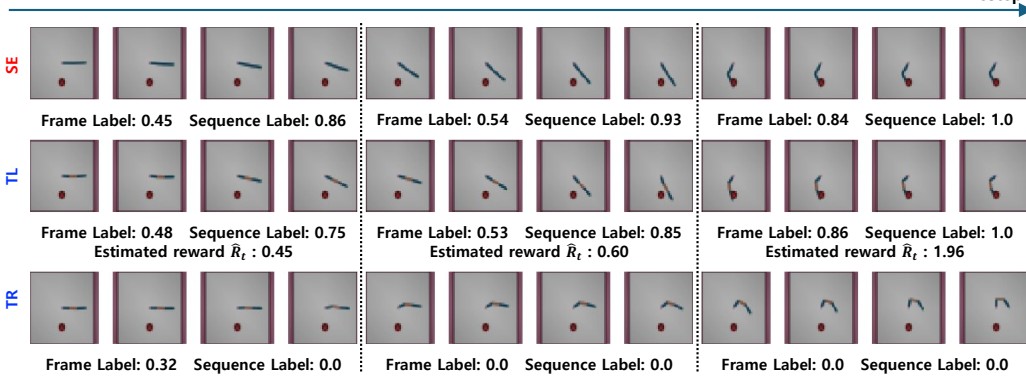

Figure E.3: Image mapping and reward analysis on RE2-to-RE3 task (Goal 2)

- **IP-to-IDP task**: Figure E.4 illustrates the IP-to-IDP task, highlighting image mapping, label estimation, and reward analysis. In this task, the pole starts upright, and the goal is to maintain balance throughout the episode. The TL data effectively aligns with SE frames across timesteps, demonstrating successful learning and accurate domain-invariant feature extraction. Initially, TR samples receive similar rewards to TL due to comparable states, but as the pole begins to fall, frame

($F_{\text{label},f}$) and sequence ($F_{\text{label},s}$) label values for TR rapidly decline to near zero. This analysis validates the proposed method's ability to reward expert-like behaviors and penalize deviations, effectively enabling robust learning in complex tasks.

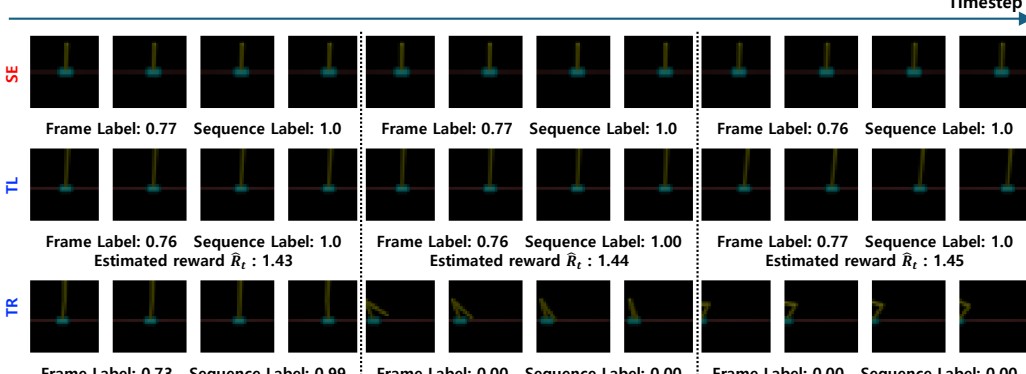

Figure E.4: Image mapping and reward analysis on IP-to-IDP task

- **DMC Pendulum task**: Figures E.5 and E.6 present the DMC Pend-to-CS and Pend-to-Acrobot tasks, highlighting the mapping and reward estimation in these challenging Pendulum environments. Initially, both frame and sequence label values are low, but they progressively increase as the agent approaches the goal, resulting in higher reward estimates. The rightmost images depict states aligned with the expert's goal, corresponding to the highest reward estimates. The frame label does not reach 1 even after achieving the goal. This occurs because the goal state is maintained until the episode ends, leading the frame label to represent an average label estimation during this period.

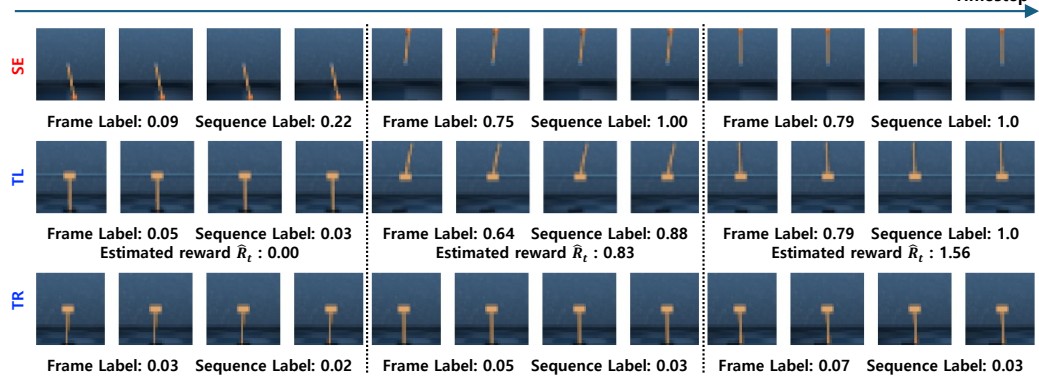

Figure E.5: Image mapping and reward analysis on Pend-to-CS task

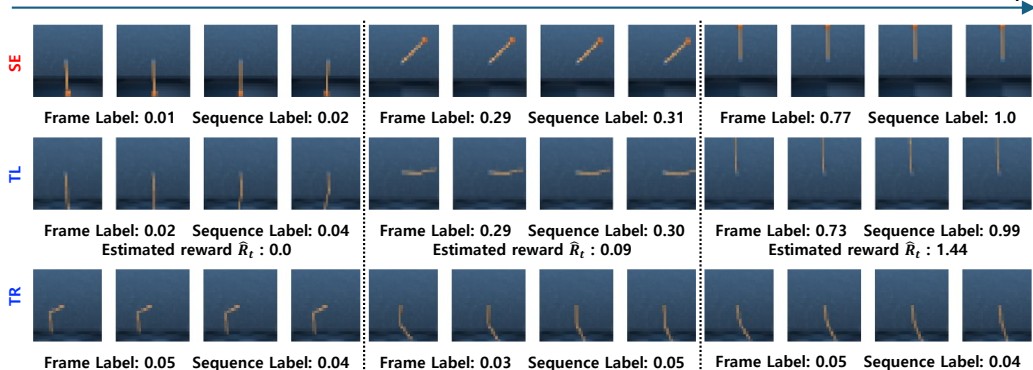

Figure E.6: Image mapping and reward analysis on Pend-to-Acrobot task

**MuJoCo Tasks**

- **Walker-to-Cheetah task**: Fig. E.7 illustrates the Walker-to-Cheetah task. At the episode's early stages, leftmost samples show label estimations and rewards assigned to observations diverging from random data. As the agent progresses, frame label predictions gradually increase, leading to higher reward estimates. By the end of the episode, both frame and sequence labels converge to values near 1, resulting in the highest reward estimates.

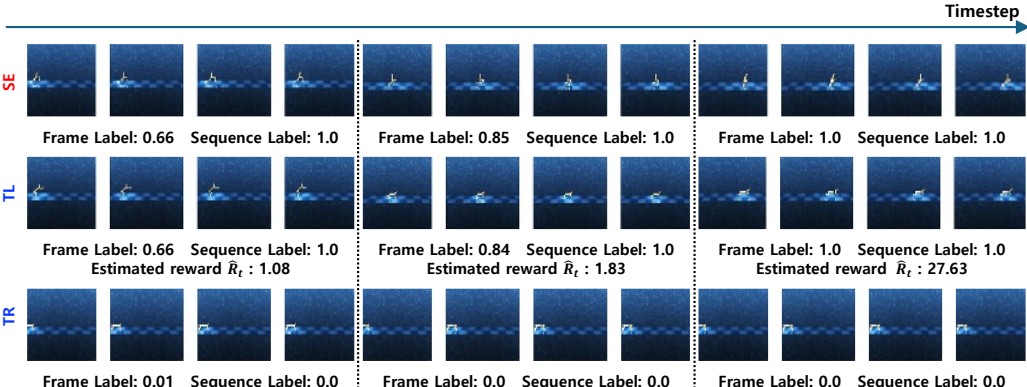

Figure E.7: Image mapping and reward analysis on Walker-to-Cheetah task

- **Walker-to-Hopper task**: In Fig. E.8, the Walker-to-Hopper task demonstrates similar trends, with frame labels and rewards increasing as timesteps progress. Unlike conventional settings where rewards are based on maintaining torso stability, our approach rewards forward velocity, prioritizing forward movement over balance. This adjustment enables the agent to achieve positions comparable to the Hopper expert by focusing on efficient locomotion.

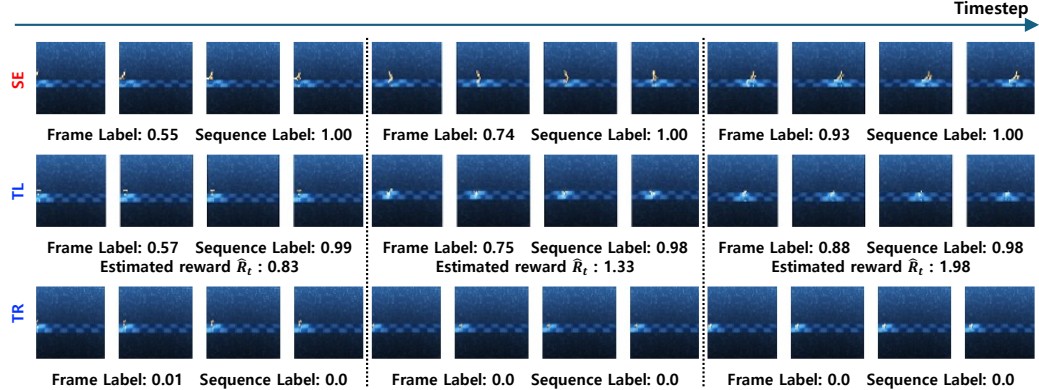

Figure E.8: Image mapping and reward analysis on Walker-to-Hopper task

- **Hopper-to-Cheetah task**: As shown in Fig. E.9, the frame label predictions in the Hopper-to-Cheetah task approach a value near 1 midway through the episode. This is influenced by the low velocity of the Hopper expert, which limits the forward distance achieved in the source domain. As a result, the target domain's performance is constrained by the Hopper's limitations, highlighting the challenges of domain adaptation in such cases.

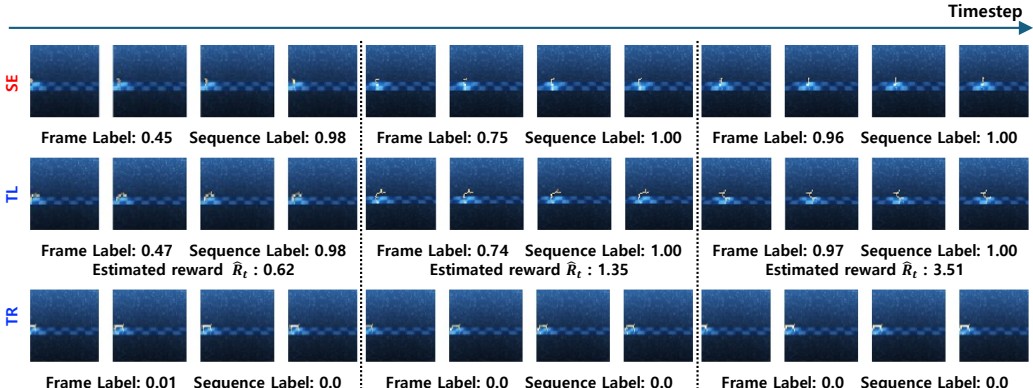

Figure E.9: Image mapping and reward analysis on Hopper-to-Cheetah task

**Robot Manipulation Tasks**

- **Pusher:R-to-H task**: As shown in Fig. E.10, both the frame and label predictions are low at the initial observations. However, starting from the middle of the episode-where there is a clear distinction from random data-the frame label values begin to increase. Notably, as the object approaches the goal, the frame label predictions remain high, indicating successful task progression.

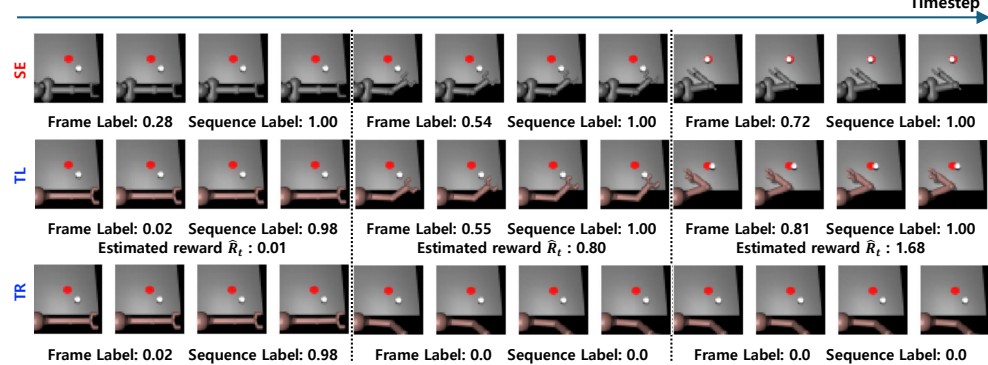

Figure E.10: Image mapping and reward analysis on Pusher:R-to-H task

- **Reach:H-to-R task**: As shown in Fig. E.11, the initial frame and sequence label values are low, similar to the Pusher environment. However, as the timesteps progress, the learner data becomes increasingly aligned with expert-like trajectories, resulting in a gradual increase in the frame label values. This demonstrates that DIFF-IL is capable of making meaningful label predictions that effectively mimic expert behavior, even in real-world-like environments.

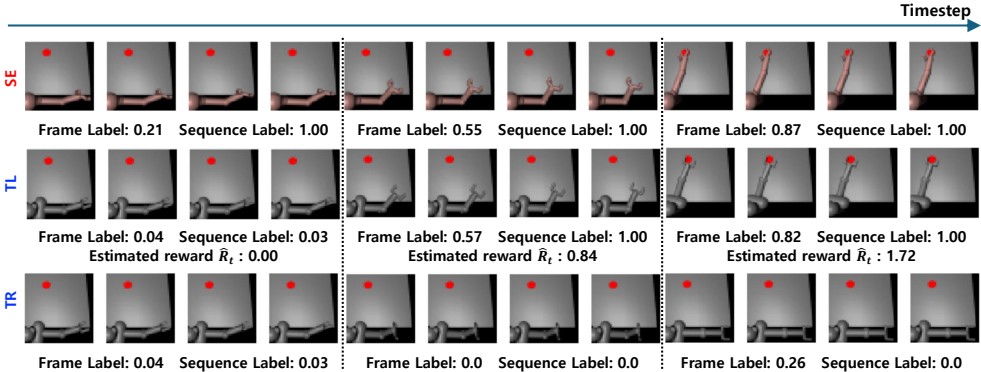

Figure E.11: Image mapping and reward analysis on Reach:H-to-R task

- **Pusher-Res:H-to-R task**: This experiment demonstrates DIFF-IL's robustness under resolution shift conditions, where the source domain images have lower resolution than the target domain. As shown in Fig. E.12, even though the source data appears blurred compared to the standard Pusher environment, DIFF-IL successfully maintains domain-invariant feature alignment throughout the episode. Both the frame and label predictions start low at initial observations but show clear improvement as the episode progresses. Starting from the middle of the episode-where there is a clear distinction from random data-the frame label values begin to increase. Notably, as the object approaches the goal, the frame label predictions remain consistently high, indicating successful task progression despite the visual degradation in source observations. This validates DIFF-IL's ability to handle practical deployment scenarios where visual fidelity may vary between training and testing environments.

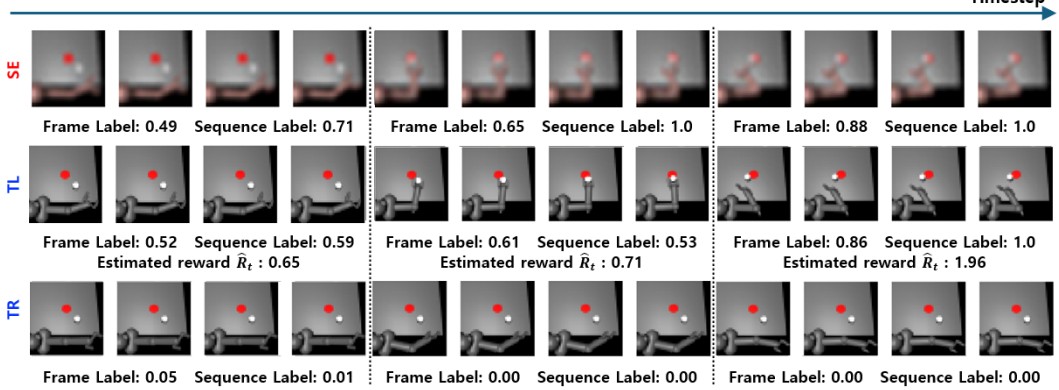

Figure E.12: Image mapping and reward analysis on Pusher-Res:H-to-R task

## F    MORE ABLATION STUDIES

This section presents an ablation study on the discriminator coefficient $\lambda_{\text{disc}}$, generator coefficient $\lambda_{\text{gen}}$, and the WGAN control coefficient $\alpha$, which significantly influence performance but were not fully detailed in the main text. In the Pendulum tasks, we conduct parameter sweeps for IP-to-IDP, RE2-to-RE3, Pend-to-CS, and Pend-to-Acrobot. Similarly, in the MuJoCo tasks, parameter sweeps are performed for Walker-to-Cheetah and Hopper-to-Cheetah. The results of these experiments are analyzed to evaluate the impact of key parameters on performance across different task environments.

### F.1    WGAN DISCRIMIANTOR LOSS COEFFICIENT $\lambda_{\text{DISC}}$

The WGAN discriminator loss coefficient $\lambda_{\text{disc}}$ controls the separation of source and target domain features. For Pendulum tasks, we swept $\lambda_{\text{disc}}$ from 1 to 50, and for MuJoCo tasks, from 0.05 to 1, reflecting the greater visual similarity in Pendulum that requires stronger discrimination. As shown in Figs. F.1 and F.2, most tasks are robust to $\lambda_{\text{disc}}$, except for IP-to-IDP (Pendulum) and Walker-to-Cheetah (MuJoCo), which show clear optimal values. These results highlight the importance of balancing feature separation: excessively high $\lambda_{\text{disc}}$ can prevent effective domain alignment, while too low values lead to uninformative features. Therefore, careful tuning of $\lambda_{\text{disc}}$ is essential, especially for tasks with challenging feature alignment.

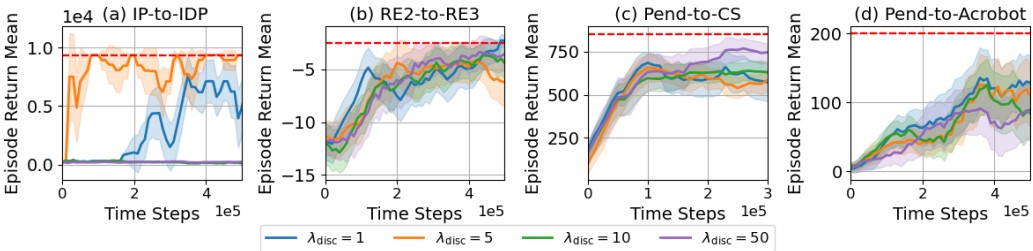

Figure F.1: Impact of the WGAN discriminator loss coefficient $\lambda_{\text{disc}}$ on Pendulum tasks

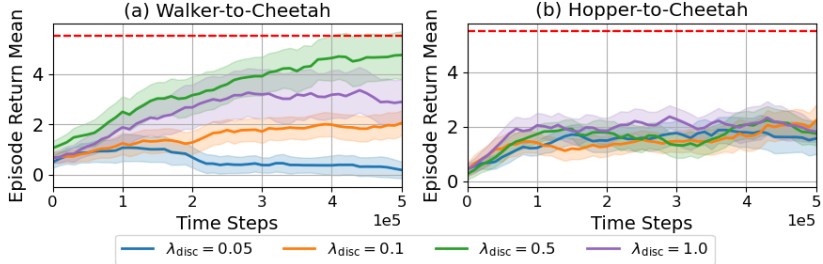

Figure F.2: Impact of the WGAN discriminator loss coefficient $\lambda_{\text{disc}}$ on MuJoCo tasks

### F.2    WGAN GENERATOR LOSS COEFFICIENT $\lambda_{\text{GEN}}$

The WGAN generator loss coefficient $\lambda_{\text{gen}}$ encourages the encoder and decoder to reduce domain feature distinguishability, with higher values promoting stronger alignment and lower values preserving domain differences. Reflecting the broader task diversity in Pendulum, we searched $\lambda_{\text{gen}}$ over $[0.05, 10]$ for Pendulum tasks and $[0.01, 1]$ for MuJoCo tasks. As shown in Figs. F.3 and F.4, most tasks exhibit robustness to $\lambda_{\text{gen}}$, except IP-to-IDP and RE2-to-RE3, where excessive alignment from high values degrades performance by obscuring critical distinctions. Conversely, Pend-to-Acrobot and Walker-to-Cheetah benefit from higher $\lambda_{\text{gen}}$, as their larger domain gaps demand aggressive alignment. These results underscore the necessity of balancing alignment strength with domain characteristics, highlighting that careful tuning of $\lambda_{\text{gen}}$-like its discriminator counterpart-is pivotal for optimal adaptation.

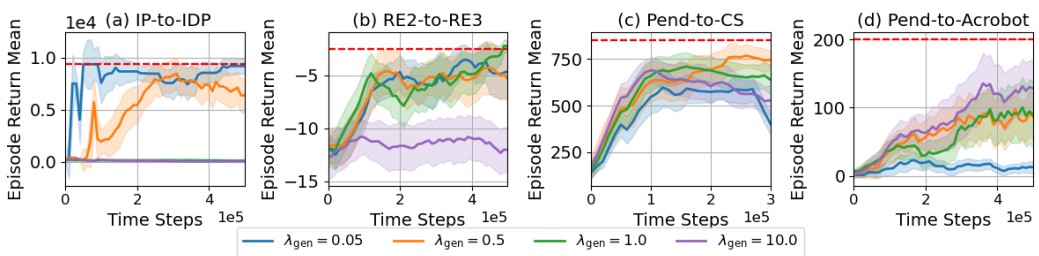

Figure F.3: Impact of the WGAN generator loss coefficient $\lambda_{\text{gen}}$ on Pendulum tasks

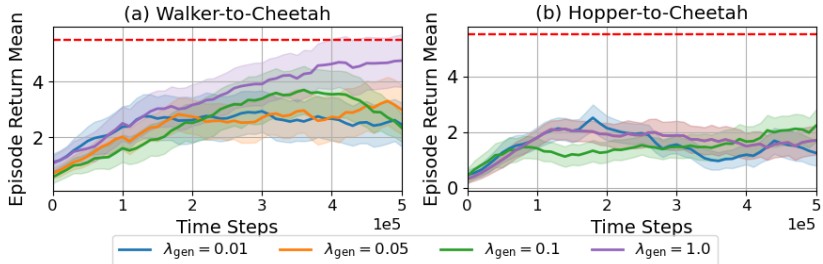

Figure F.4: Impact of the WGAN generator loss coefficient $\lambda_{\text{gen}}$ on MuJoCo tasks

## F.3 WGAN CONTROL COEFFICIENT $\alpha$

We performed an extended search for the WGAN control coefficient $\alpha$, which balances frame and sequence mapping, across both Pendulum and MuJoCo tasks at $\alpha \in [0.1, 0.5, 0.9]$. As shown in Figs. F.5 and F.6, $\alpha = 0.5$ consistently yields the best results, while prioritizing either mapping leads to performance drops. This is because DIFF-IL relies on both frame and sequence alignment for label prediction, making balanced weighting essential for effective domain-invariant feature extraction and imitation.

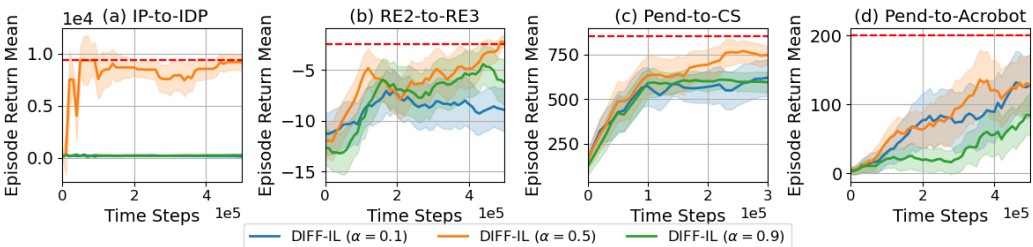

Figure F.5: Ablation study : WGAN control factor $\alpha$ on Pendulum tasks

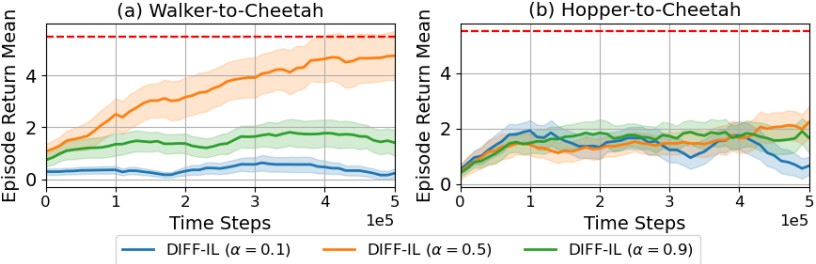

Figure F.6: Ablation study : WGAN control factor $\alpha$ on MuJoCo tasks

## G  PERFORMANCE COMPARISON WITH TCN

To provide a comprehensive evaluation, we include a comparison with the Time Contrastive Network (TCN) (Sermanet et al., 2018), a temporal correspondence-based method that differs structurally from other baselines. We implemented TCN using single-view training on source data and provided rewards based on Huber-style loss calculated between target data and corresponding timestep observations, following the original paper's default hyperparameters.

| Task | Pend-to-Acrobot | Walker-to-Cheetah | Pusher:H-to-R | Reach:H-to-R |
|------|-----------------|-------------------|---------------|--------------|
| DIFF-IL | $\mathbf{128.24 \pm 40.58}$ | $\mathbf{4.54 \pm 0.86}$ | $\mathbf{-52.79 \pm 17.78}$ | $\mathbf{-137.10 \pm 9.20}$ |
| TCN | $4.51 \pm 3.61$ | $-0.15 \pm 0.54$ | $-105.35 \pm 0.57$ | $-335.50 \pm 58.99$ |

Table G.1: Performance comparison between DIFF-IL and TCN across selected cross-domain tasks.

Table G.1 shows the performance comparison across four representative cross-domain scenarios. TCN's reliance on MSE-based timestep matching presents challenges in environments where the temporal dynamics for goal achievement vary significantly between domains. The method assigns rewards based on current timestep correspondence, which becomes problematic when agents exhibit different goal achievement patterns due to varying physical properties across domains. Additionally, without explicit domain confusion mechanisms, TCN struggles to bridge domain gaps effectively, resulting in suboptimal reward estimation even in scenarios with similar temporal requirements. These limitations lead to performance comparable to other baselines that face similar challenges in our cross-domain settings, further validating the necessity of DIFF-IL's domain-invariant feature extraction and frame-wise temporal labeling approaches.

