# OpenReview forum: "Domain-Invariant Per-Frame Feature Extraction for Cross-Domain Imitation Learning with Visual Observations"
_ICLR.cc/2026/Conference — ICLR 2026 Conference Withdrawn Submission_

### Official Review · Reviewer_Cr1Z · 2025-10-18

**Soundness:** 2
**Presentation:** 3
**Contribution:** 2
**Rating:** 4
**Confidence:** 4

**Summary:**

The paper introduces DIFF-IL, a method for cross-domain imitation learning from visual observations. It extracts domain-invariant per-frame features with a WGAN-based encoder and decoder, removes residual domain cues at the sequence level, and applies a frame-wise time-labeling reward that emphasizes later timesteps. Evaluations on Pendulum, MuJoCo locomotion, robot manipulation with embodiment and resolution shifts show consistent improvements over recent baselines.

**Strengths:**

- The paper is easy to follow, with a clear presentation of the motivation and method.
- The problem is important, addressing key challenges in transferring skills across different embodiments and visual modalities.
- Strong baselines and experimental results are provided within the context of cross-domain imitation learning.

**Weaknesses:**

- The framework combines known ideas such as domain confusion, reconstruction, and time-label rewards (temporal distance prediction to the final goal) [1] though it does not introduce a fundamentally new learning principle.
- The experimental assumptions are somewhat restrictive. In the imitation learning settings, the goal’s visual features are assumed to be similar across domains, and evaluations are restricted to such environments.
- Moreover, there is limited analysis of source-target similarity, making it unclear how the framework behaves when the domains differ significantly.
- According to Table 1 and Figure 8, the performance gain of DIFF-IL heavily relies on the inclusion of the time-label component, as shown by the clear drop in the $W/O \ F_\{\text{label,f}}$ ablation. However, because the paper lacks a detailed analysis of the time-label design itself (for example, the effect of the label-loss weight $\lambda_{\text{label,f}}$ or the choice of label-value range), it remains unclear how sensitive the method is to these design decisions, which could limit the robustness and generality of the approach.
- The evaluated environments use simple reward designs, mostly distance-based. Since target policies are trained online through simulated interaction, a comparison with an oracle baseline (for example, SAC trained directly on target rewards) would clarify the degree of success achieved by the proposed method.
- It is also unclear how well this framework generalizes to more complex or less visually aligned cross-domain settings.

[1] Aytar, Yusuf, et al. "Playing hard exploration games by watching youtube." Advances in neural information processing systems 31 (2018).

**Questions:**

- In Figure 5, what would happen if the source cheetah’s motion direction were opposite to the target walker’s? How could DIFF-IL be extended to handle such cases? Similarly, how would it deal with vertical inversion in the Pendulum environment or symmetry changes in manipulation tasks?
- How robust is the method to visual noise or perturbations in the target observations?
- In Section 5.3, the statement “…whereas competing methods often fail to replicate expert performance” seems quite strong. Could it be clarified whether, under the experimental settings, the combination of the source expert, randomly sampled source trajectory and randomly sampled target trajectories alone is sufficient to reproduce the target policy?

---

### Official Review · Reviewer_gRjM · 2025-10-29

**Soundness:** 3
**Presentation:** 2
**Contribution:** 2
**Rating:** 4
**Confidence:** 4

**Summary:**

In this paper, the authors propose a new method, DIFF-IL (Domain-Invariant Per-Frame Feature Extraction for Imitation Learning), to address the challenges of high-dimensional, noisy, and incomplete visual observations in Cross-Domain Imitation Learning (CDIL). Its core contributions include: (1) extracting domain-invariant features frame by frame, removing domain-specific information by sharing encoder and a domain-specific decoder structure with Wasserstein Gan, while preserving task-related behavioral details; (2) the frame-wise time labeling mechanism is introduced to label the expert trajectories in fine-grained time steps, and guide the learner to preferentially imitate the later behavior close to the task target. Experiments on MuJoCo, DMC, and robot manipulation tasks verify the method's superior performance across differences in morphology, dynamics, and resolution.

**Strengths:**

- This paper focuses on single-frame domain invariant feature extraction, which effectively alleviates the problem of alignment failure caused by differences in view angle, shape, and resolution, and has clear application value.
- The WGAN is used for frame-by-frame domain alignment, and the sequence-level and frame-level dual discriminators are combined to remove domain-specific information while preserving task semantics. In addition, the time labeling mechanism does not simply reuse TCN's time alignment idea; it translates it into a learnable reward signal that is naturally fused with the adversarial imitation learning framework, thereby improving learning efficiency. It embodies the ingenious integration and improvement of the existing technology.
- The paper constructed 22 cross-domain migration scenarios on three categories of tasks, covering challenges such as morphological differences (e.g., Walker → Cheetah), resolution changes (- res variant), etc.. Compared with SOTA methods, DIFF-IL is significantly superior across most tasks and provides multi-dimensional verification.

**Weaknesses:**

- Although the frame-level time label (yt = (t / H + 1)/2) is effective in experiments, its design is more heuristic and lacks a theoretical connection with task structure or optimal strategy. For example, why are linear growth labels better than exponential or other forms? Is it suitable for non-monotonous tasks (e.g., moving away from a goal before approaching it)? It is suggested to supplement the sensitivity analysis or theoretical basis for the form of the label function.
- TCN is used as a baseline for comparison, but it is not specifically designed for imitative learning, and its single-view training setting may not be fully exploited. More critically, recent examples such as XIRL (Zakka et al., 2022), D3IL (Choi et al., 2024), and others have also employed temporal consistency or dual encoders, which can be used to encode a single signal, however, DIFF-IL does not clearly clarify its essential difference from these methods in feature decoupling mechanism or reward design, which leads to the question of Incremental improvement.
- Although Appendix D mentions that DIFF-IL training is faster than D3IL, it introduces two discriminators (frames + sequences) + two labeling networks + two decoders, with significantly higher model complexity than TPIL or DeGAIL. In real robotic systems, inference latency and memory footprint are crucial, yet papers do not provide computational overhead metrics (e.g., FLOPs, latency) or lightweight inference strategies, weakening their appeal for real deployments.

**Questions:**

- Are linear time labels still valid in non-monotonic tasks, such as “Bypassing obstacles before reaching the target”? Have you considered a dynamic label generation mechanism based on state value (such as $V(s_t)$) or subgoal progress?
- The paper emphasizes the removal of domain-specific information, but some domain differences, such as cheetahs running faster than walkers, themselves contain task-related dynamic information. How can DIFF-IL ensure that the dynamic nature of such tasks is not lost while dedomaining? Are there quantitative metrics (e. g. MMD, CORAL loss) that measure the trade-off between domain alignment and task information retention?
- D3IL also uses double consistency (cycle consistency + reconstruction) to extract behavioral characteristics. What are the fundamental advantages of DIFF-IL's “frame-by-domain invariant features + time labels” over D3IL's “dual encoders + consistency constraints” in terms of feature representation ability or reward signal quality? Can finer-grained ablation be performed in the same experimental setting (e.g. , fixed feature extraction module, replacing only the reward mechanism) ?

---

### Official Review · Reviewer_Nccm · 2025-10-29

**Soundness:** 3
**Presentation:** 2
**Contribution:** 3
**Rating:** 4
**Confidence:** 4

**Summary:**

This paper focuses on cross-domain imitation learning from visual observations. An imitation learning method is proposed to extract domain-invariant features from frames individually and then fine-tune them over sequences to isolate and replicate the expert’s behaviour. Additionally, a frame-wise labelling technique is proposed to segment expert behaviours and assign rewards aligned with the temporal context. The proposed approach, DIFF-IL, is evaluated and compared with existing approaches in implemented MuJoCo environments. The results show that the proposed approach offers benefits compared to the other methods.

**Strengths:**

The sequence matching, frame-wise labelling, and reward design components appear to be novel contributions within the context of imitation learning from visual observations in simulated environments.

The proposed methods are thoroughly evaluated in the developed simulation settings, providing strong empirical support for the approach.

The paper is well written and the authors already shared their code, demonstrating their commitment to transparency.

**Weaknesses:**

The paper is clearly written and easy to follow. However, there are several weaknesses regarding the contribution, novelty, and potential impact of the work. While the topic is relevant and the presentation is solid, the paper does not provide sufficient advancement beyond existing methods.

A key concern is the limited experimental setup. After several years of progress in this area, the work still relies primarily on the same MuJoCo environments used in earlier studies, with only minor variations (e.g., resolution shifts). Although the paper refers to a “dexterous multi-fingered hand (humanoid)” setup, no experiment demonstrates this scenario. As a result, the broader impact and practical relevance of the proposed approach remain unclear.

The method transfers knowledge from third-person view to third-person view, which limits its applicability to real-world robot learning contexts. Compared to approaches such as TCN, the connection to physical or real-robot implementations is not well established. It remains uncertain whether the method could generalise beyond the simulated environments, making the overall contribution appear incremental. Beyond outperforming a few prior baselines, the paper does not demonstrate additional benefits or insights.
The experimental results are not presented in a way that allows assessment of robustness across different factors, such as embodiment, background variation, or camera viewpoint. The visualised examples mainly illustrate differences in embodiment, which restricts the interpretability of the findings.

Furthermore, the motivation for the frame-level approach is insufficiently justified. For instance, previous work such as DeGAIL utilised sequential inputs to capture task progression. In contrast, DIFF-IL introduces an additional stage that assigns higher labels to later timesteps, but this design choice is not clearly motivated or empirically supported in the paper.

Overall, while the paper is well written and addresses an interesting problem, the novelty and general impact are limited. Substantial extensions to the experimental setup, clearer motivation for methodological choices, and stronger evidence of real-world applicability would be required for the work to make a significant contribution to the field.

**Questions:**

1) Please clarify how the experimental setup extends beyond the works introduced a few years ago. In its current form, it is not clear what new challenges or environments are being addressed compared to previous studies.
2) Please justify the need for combining a frame-level and sequence-level learning approach. What specific advantage does this hybrid formulation provide over sequential models?
3) Please clarify the practical relevance of the proposed method. How could this approach have an impact beyond the current simulated environments, particularly in real-world or robotic applications?
4) Please elaborate on how this work addresses the limitations of prior studies, beyond simply improving performance on existing benchmarks introduced several years ago. What new insights does it offer to the field?

---

### Official Review · Reviewer_1Bsu · 2025-10-31

**Soundness:** 2
**Presentation:** 2
**Contribution:** 2
**Rating:** 2
**Confidence:** 4

**Summary:**

This paper aims to tackle the challenge of cross-domain imitation learning (CDIL) from visual observations. The authors argue that existing methods, which often rely on aligning sequences of images, struggle with complexity and misalignment. The paper proposes Domain-Invariant Per-Frame Feature Extraction and introduces a time-labeled reward as an additional reward design. Experiments on Pendulum, MuJoCo, and new Robot Manipulation tasks show that DIFF-IL outperforms existing visual CDIL baselines.

**Strengths:**

The primary strength of the paper is its strong empirical results. DIFF-IL outperforms baselines across a wide range of tasks and different cross-domain settings. The ablation study provides clear evidence for the importance of the method's components.

**Weaknesses:**

1. The paper's primary premise and title are "Per-Frame Feature Extraction". It is motivated against sequence-based methods, claiming they lead to "misaligned features" and "suboptimal imitation". However, the proposed DIFF-IL method explicitly relies on sequences. The paper's core contribution is not "per-frame" vs. "sequence," but rather a multi-level alignment (first frame, then sequence).
2. The contribution is limited. The domain-invariant feature extraction (DIFF) component is a combination of WGANs , autoencoders , and feature consistency loss. The multi-level (frame-then-sequence) WGAN alignment is a reasonable but incremental extension.
3. The most novel part of the method appears to be the "frame-wise time label". This is a hard-coded heuristic that assumes expertness is directly proportional to the timestep. This heuristic is well-suited for the chosen tasks (locomotion: "move forward"), which inherently have a reward correlated with time/survival, but would fail on any task without simple, monotonic progress.
4. The line spacing in the paper is too narrow and highly inconsistent with the template, notably affecting the subtitles, figures, tables and captions, and their separation from the main body text.

**Questions:**

The time reward is based on the assumption that expertness is proportional to the timestep. How would this method perform on tasks that are not simple monotonic progress? For example, a task requiring the agent to move to location A, then location B, and then back to location A? Wouldn't the time-label heuristic fail completely here, as it would just reward being at the final location?

---

### Note · Authors · 2025-11-14

I have read and agree with the venue's withdrawal policy on behalf of myself and my co-authors.